# Active Bipartite Ranking

**James Cheshire**      **Stephan Clemencon**
Telecom ParisTech
first.last@telecom-paris.fr

**Vincent Laurent**
ENS Paris Saclay
first.last@ens-paris-saclay.fr

## Abstract

In this paper, we develop an active learning framework for the bipartite ranking problem. Motivated by numerous applications, ranging from supervised anomaly detection to credit-scoring through the design of medical diagnosis support systems, and usually formulated as the problem of optimizing (a scalar summary of) the ROC curve, bipartite ranking has been the subject of much attention in the passive context. Various dedicated algorithms have been recently proposed and studied by the machine-learning community. In contrast, active bipartite ranking rule is poorly documented in the literature. Due to its global nature, a strategy for labeling sequentially data points that are difficult to rank w.r.t. to the others is required. This learning task is much more complex than binary classification, for which many active algorithms have been designed. It is the goal of this article to provide a rigorous formulation of such a selective sampling approach. We propose a dedicated algorithm, referred to as `active-rank`, which aims to minimise the distance between the ROC curve of the ranking function built and the optimal one, w.r.t. the sup norm. We show that, for a fixed confidence level $\varepsilon$ and probability $\delta$, `active-rank` is PAC($\varepsilon, \delta$). In addition, we provide a problem dependent upper bound on the expected sampling time of `active-rank` and also demonstrate a problem dependent lower bound on the expected sampling time of any PAC($\varepsilon, \delta$) algorithm. Beyond the theoretical analysis carried out, numerical results are presented, providing strong empirical evidence of the performance of the algorithm proposed, which compares favorably with more naive approaches.

## 1 Introduction

In bipartite ranking, the statistical framework is exactly the same as that in standard binary classification, the flagship problem in statistical learning theory. One observes $n \geq 1$ independent copies $\mathcal{D}_n = \{(X_1, Y_1), \ldots, (X_n, Y_n)\}$ of a generic random pair $(X, Y)$ with (unknown) distribution $P$, where $Y$ is a binary random label, valued in $\{-1, +1\}$ say, and $X$ is a high dimensional random vector, taking its values in $\mathcal{X} \subset \mathbb{R}^d$ with $d \geq 1$, that models some information hopefully useful to predict $Y$. In contrast to binary classification, the goal pursued is of global (and not local) nature. It is not to assign a label, positive or negative, to any new input observation $X$ but to rank any new set of (temporarily unlabeled) observations $X'_1, \ldots, X'_{n'}$ by means of a (measurable) scoring function $s : \mathcal{X} \to \mathbb{R}$, so that those with positive label appear on top of the list (*i.e.* are those with the highest scores) with high probability. More formally, the accuracy of any scoring rule can be evaluated through the ROC curve criterion or its popular scalar summary, the AUC (standing for the Area Under the ROC Curve), and, as expected, optimal scoring functions w.r.t. these performance measures can be shown to be increasing transforms of the posterior probability $\eta(x) = \mathbb{P}\{Y = +1 \mid X = x\}$,

$x \in \mathcal{X}$. Though easy to formulate, this problem encompasses many applications, ranging from credit risk screening to the design of decision support tools for medical diagnosis through (supervised) anomaly detection. Hence, motivated by a wide variety of applications, bipartite ranking has received much attention these last few years. Many approaches to this global learning problem (*i.e.* the problem of learning a preorder on the input space $\mathcal{X}$ based on a binary feedback) have been proposed and investigated. In Clémençon and Vayatis (2009), optimization of the (empirical) ROC curve in sup norm is considered via nonlinear approximation techniques, while this functional optimization problem is viewed as a superposition of cost-sensitive binary classification problems in Clémençon and Vayatis (2008) so as to propose an alternative method. In Clémençon et al. (2008) (see also Agarwal et al. (2005)), empirical maximization of the empirical AUC criterion is considered, bipartite ranking being viewed as a pairwise classification problem, a *plug-in* approach to bipartite ranking is developed in Clémençon and Robbiano (2011) and scalar performance criteria other than the AUC have been recently reviewed in Menon and Williamsson (2016). Whereas the vast majority of dedicated articles consider the *batch* situation solely, where the learning procedure fully relies on a set $\mathcal{D}_n$ of training examples given in advance, the goal of this paper is to develop an *active learning* framework for bipartite ranking, in other words to investigate this problem in an iterative context, where the learning procedure can formulate queries in a sequential manner, so as to observe the labels at new data points in order to refine progressively the scoring/ranking model. Precisely, the challenge consists in determining an incremental experimental design to label the data points in $\mathcal{X}$ that would permit to improve the ROC curve progressively, with statistical guarantees.

**Our contributions**  We describe an algorithm, `active-rank`, which sequentially queries points of the feature space $\mathcal{X}$. Given a confidence level $\varepsilon > 0$ and probability $\delta > 0$, the goal of `active-rank` is, in a few queries as possible, to output a ranking of $\mathcal{X}$, such that the induced ROC curve of said ranking is within $\varepsilon$ of the optimal ROC curve, in terms of the sup norm, with probability greater than $1 - \delta$. We restrict our selves to dimension 1, i.e. $\mathcal{X} = [0, 1]$ and make a single key assumption that the posterior $\eta$ is piecewise constant on a grid of size $K$. Theorem 3.2 then shows that `active-rank` satisfies the above statistical guarantee and furthermore, provides an upper bound on it's expected total number of queries. In Theorem 3.3 we provide a lower bound on the expected number of queries for any possible policy, which satisfies a confidence level $\varepsilon$ with probability greater than $1 - \delta$. The aforementioned bounds are *problem dependent*, in the sense that they depend on features of the posterior $\eta$. Finally we conduct a practical analysis of `active-rank` on synthetic data, comparing it to several naive approaches.

The article is structured as follows. In Section 2 we formally define our setting as well recalling some key notions related to bipartite ranking and ROC analysis. In Section 2 we also cover some of the existing literature in active learning, of relevance to bipartite ranking. We then describe the `active-rank` algorithm in Section 3.1. Following this our theoretical results are presented in Section 3.2. Lastly, the experiments are displayed in Section 4. We then conclude and discuss some perspectives for future research in Section 5. Technical details and proofs are deferred to the Supplementary Material.

## 2   Background and preliminaries

### 2.1   Notation

Here we introduce several dedicated notions that will be extensively used in the subsequent analysis. For any integer $n \geq 1$, we set $[n] := \{1, \ldots, n\}$, denote by $\mathfrak{S}_n$ the symmetric group of permutations on $\{1, \ldots, n\}$, by $\mathbf{I}_n$ the identity map of $\mathfrak{S}_n$. By $\lambda$ is meant the Lebesgue measure on $[0, 1]$. Given two probability distributions $P$ and $Q$ on a measurable space $(\Omega, \mathcal{F})$, we write $P \ll Q$ when $P$ is absolutely continuous w.r.t. $Q$. For any $a, b$ in $[0, 1]$, $\mathcal{B}\mathrm{er}(a)$ refers to the Bernoulli distribution with mean $a$ and $\mathrm{kl}(a, b)$ to the Kullback Leibler divergence between the Bernoulli distributions $\mathcal{B}\mathrm{er}(a)$ and $\mathcal{B}\mathrm{er}(b)$. For any $a, b \in [0, 1]$, the Chernoff Information between the distributions $\mathcal{B}\mathrm{er}(a)$ and $\mathcal{B}\mathrm{er}(b)$ is defined as, $\mathrm{kl}^*(a, b) = \mathrm{kl}(x^*, a) = \mathrm{kl}(x^*, b)$, where $x^*$ is the unique $x \in [0, 1]$ such that $\mathrm{kl}(x, a) = \mathrm{kl}(x, b)$. The indicator function of any event $\mathcal{E}$ is denoted by $\mathbb{I}\{\mathcal{E}\}$, the Dirac mass at any point $x$ by $\delta_x$, and the pseudo-inverse of any cdf $\kappa(u)$ on $\mathbb{R}$ by $\kappa^{-1}(t) = \inf\{v \in \mathbb{R} : \kappa(v) \geq t\}$.

## 2.2 Setting

**The bipartite ranking problem** A rigorous formulation of bipartite ranking involves functional performance measures. Let $\mathcal{S}$ be the set of all scoring functions, any $s \in \mathcal{S}$ defines a preorder $\preceq_s$ on $\mathcal{X}$: for all $(x, x') \in \mathcal{X}^2$, $x \preceq_s x' \Leftrightarrow s(x) \leq s(x')$. From a quantitative perspective, the accuracy of any scoring rule can be evaluated through the ROC curve criterion, namely the PP-plot $t \in \mathbb{R} \mapsto (1 - H_s(t), \ 1 - G_s(t))$, where $H_s(t) = \mathbb{P}\{s(X) \leq t \mid Y = -1\}$ and $G_s(t) = \mathbb{P}\{s(X) \leq t \mid Y = +1\}$, for all $t \in \mathbb{R}$. The curve can also be viewed as the graph of the càd-làg function $\alpha \in (0, 1) \mapsto \mathrm{ROC}(s, \alpha) = 1 - G_s \circ H_s^{-1}(1 - \alpha)$. The notion of ROC curve defines a partial order on the set of all scoring functions (respectively, the set of all preorders on $\mathcal{X}$): $s_1$ is more accurate than $s_2$ when $\mathrm{ROC}(s_2, \alpha) \leq \mathrm{ROC}(s_1, \alpha)$ for all $\alpha \in (0, 1)$. As can be proved by a straightforward Neyman-Pearson argument, the set $\mathcal{S}^*$ of optimal scoring functions is composed of increasing transforms of the posterior probability $\eta(x) = \mathbb{P}\{Y = +1 \mid X = x\}$, $x \in \mathcal{X}$. We have $\mathcal{S}^* = \{s \in \mathcal{S} : \ \forall(x, \ x') \in \mathcal{X}^2, \ \eta(x) < \eta(x') \Rightarrow s^*(x) < s^*(x')\}$ and

$$\forall(s, s^*) \in \mathcal{S} \times \mathcal{S}^*, \ \forall \alpha \in (0, 1), \ \ \mathrm{ROC}(s, \alpha) \leq \mathrm{ROC}^*(\alpha) := \mathrm{ROC}(s^*, \alpha).$$

The ranking performance of a candidate $s \in \mathcal{S}$ can be thus measured by the distance in sup-norm between its ROC curve and $\mathrm{ROC}^*$, namely $d_\infty(s, s^*) := \sup_{\alpha \in (0,1)}\{\mathrm{ROC}^*(\alpha) - \mathrm{ROC}(s, \alpha)\}$ . An alternative convention to represent the ROC of a scoring function $s$, which we will use for the remainder of this paper, is to consider the broken line $\widetilde{\mathrm{ROC}}(s, .)$, which arises from connecting the PP-plot by line segments at each possible jump of the cdf $H_s$. **From here out when referring to the ROC of a scoring function $s$, we refer to the broken line $\widetilde{\mathrm{ROC}}(s, .)$.**

**The active learning setting** Whereas in the batch mode, the construction of a nearly optimal scoring function (*i.e.* a function $s \in \mathcal{S}$ such that $d_\infty(s, s^*)$ is 'small' with high probability) is based on a collection of independent training examples given in advance, the objective of an *active learner* is to formulate queries in order to recover sequentially the optimal preorder on the feature space $\mathcal{X}$ defined by the supposedly unknown function $\eta$. That is, the active learner plays a game with multiple time steps, where, at time each step $n$, they must choose a point $a_n \in \mathcal{X}$ to query, so as to observe the random label $Y_n \sim \mathcal{B}\mathrm{er}(\eta(a_n))$ and refine the scoring model incrementally. After a sufficient number of rounds has elapsed, chosen at the learner's discretion, a final scoring function $\hat{s}$, is output.

**Piecewise constant scoring functions.** Here we consider the simplest scoring functions, measurable functions that are constant on pieces of the input space $\mathcal{X}$ forming a partition. As shown in Clemencon and Vayatis (2009) (see subsection 2.3 therein), when smooth enough, $\mathrm{ROC}^*$ can be accurately approximated by the (stepwise) ROC curve of a piecewise constant scoring function. Because the goal of this paper is to highlight the nature of active bipartite ranking rather than treating the problem in full generality, various simplifying assumptions are made in the subsequent analysis. For simplicity we suppose that $\mathcal{X} = [0, 1)$ and introduce the grid points $\{G_1, ..., G_K\} = \{i/K : \ i = 1, \ ..., \ K - 1\}$, where $K \geq 1$. A preorder on $\mathcal{X}$ can be then naturally defined by means of a permutation $\sigma \in \mathfrak{S}_K$. Consider indeed the scoring function

$$s_\sigma(x) := \sum_{i=1}^{K} i \cdot \mathbb{I}\{x \in [G_{\sigma(i)}, \ G_{\sigma(i+1)})\}. \tag{1}$$

We denote by $\mathcal{S}_K$ the set of all functions of type (1). To avoid dealing with model bias here, we assume that the optimal preorder, that induced by $\eta(x)$ namely, can be defined by a scoring function in $\mathcal{S}_K$.

**Assumption 2.1.** There exist a permutation $\sigma \in \mathfrak{S}_K$ and distinct constants $\mu_1, \ ..., \ \mu_K$ in $(0, 1)$ such that

$$\eta(x) = \sum_{i=1}^{K} \mu_i \cdot \mathbb{I}\{x \in [G_{\sigma(i)}, \ G_{\sigma(i+1)})\} \text{ for all } x \in [0, 1) \ .$$

We write $p = \frac{1}{K}\sum_{i\in[K]}\mu_i$. We point out that, as $\eta$ may remain constant over multiple sections of the grid, the permutation $\sigma$ satisfying assumption 2.1, is not necessarily unique. In the subsequent analysis, the parameter $K$ is supposed to be known, in contrast with the $\mu_i$'s, which have to be learned by means of an active strategy. As we assume no structure between the grid points, one can easily consider Assumption 2.1 in higher dimensions, simply consider the $d$ dimensional grid of size $K^d$. As such, all results presented in this paper **extend trivially to higher dimensions**.

**Policies and fixed confidence regime.** We denote the outputted scoring function of the learner $\hat{s} \in S_K$. The way the learner interacts with the environment - i.e. their choice of points to query, how many samples to draw in total and their choice of $\hat{s} \in S_K$, we term the *policy* of the learner. We write $\mathcal{C}$ for the set of all possible policies of the learner. For a policy $\pi \in \mathcal{C}$ and problem $\nu \in \mathcal{B}$ we denote random variable $\tau_\nu^\pi$ as the stopping time of policy $\pi$. We write $\hat{s}_\nu^\pi$ for the scoring function outputted by policy $\pi$ on problem $\nu$. Where obvious we may drop the dependency on $\pi, \nu$ in the notation, referring to the scoring function outputted by the learner as simply $\hat{s}$. We write $\mathbb{P}_{\nu,\pi}$ as the distribution on all samples gathered by a policy $\pi$ on problem $\nu$. We similarly define $\mathbb{E}_{\nu,\pi}$.

For the duration of this paper we will work in the *fixed confidence regime*. For a confidence level $\varepsilon$, define, $S_K^\varepsilon := \{s \in S_K : d_\infty(s, \eta) \leq \varepsilon\}$. A policy $\pi$ is said to be PAC$(\delta, \varepsilon)$ (probably approximately correct), on the class of problems $\mathcal{B}$, if, $\forall \nu \in \mathcal{B}, \mathbb{P}_{\nu,\pi}[\hat{s} \in S_K^\varepsilon] \geq 1 - \delta$. The goal of the learner is to then obtain a PAC$(\delta, \varepsilon)$ policy $\pi$, such that the expected stopping time in the worst case, $\sup_{\nu \in \mathcal{B}} \mathbb{E}_{\nu,\pi}[\tau_\nu^\pi]$, is minimised.

**Defining problem complexity** The expected minimum number of samples a policy must draw on a certain problem, $\nu \in \mathcal{B}$, to be PAC$(\delta, \varepsilon)$ is a quantity which depends upon the features of $\nu$, specifically, the shape of the posterior $\eta$. When defining our measure of problem complexity we must capture this dependence as succinctly as possible. To build intuition for our definition we first explore a naive strategy and introduce some informative Lemmas. A naive approach to the active bipartite ranking problem, is to treat each pair of points on the grid, $i, j \in [K]$ as a separate classification problem. To correctly distinguish the situations, $\mathcal{H}_0^{i,j} := \mu_i > \mu_j, \mathcal{H}_1^{i,j} := \mu_i < \mu_j$, with probability greater than $1 - \delta$, it is well known, see e.g. Kaufmann et al. (2014), that for small $\delta$, the minimum number of samples required is of the order $\frac{\log(1/\delta)}{\mathrm{kl}^*(\mu_j, \mu_i)}$, where we remind the reader $\mathrm{kl}^*$ is the the Chernoff Information, closely related to the kl divergence, see Section 2.1. Thus, if the learner wished to output a scoring function in $S^*$, the sample complexity would be of the order, $\sum_{i \in [K]} \frac{\log(1/\delta)}{\min_{j \in [K]}(\mathrm{kl}^*(\mu_j, \mu_i))}$. Of course, distinguishing between $\mathcal{H}_0^{i,j}$ and $\mathcal{H}_1^{i,j}$ is impractical when $\mu_i$ and $\mu_j$ are very close, or even equal. However, in our regime, the learner is not required to correctly rank every pair of points $i, j \in [K]$, only to output a scoring function existing in $S_K^\varepsilon$. Intuition indicates that the learner may be irreverent to the ranking within certain groups of points on the gird, as long as there posterior values are sufficiently close. For instance, consider a partition of $[0, 1]$, $\mathcal{P} = \{C_1, C_2, C_3\}$, and increasing sequence $(\beta_1, \beta_2, \beta_3) \in [0, 1]^3$ with

$$\eta(x) = \sum_{i=1}^3 \mathbb{I}(x \in C_i)\beta_i, \qquad \tilde{s}(x) = \mathbb{I}(x \in C_1) + 2\mathbb{I}(x \in \{C_2 \cup C_3\}), \tag{2}$$

Where the scoring function function $\tilde{s}$ essentially, treats all points of $C_2, C_3$ of the same rank. See Figure 1 for the ROC curves of $\eta$ and $\tilde{s}$. Via simple calculation, we have the following, $d_\infty(\eta, \tilde{s}) = \frac{\lambda(C_3)}{p} \frac{(\beta_3 - \beta_2)/2}{1 - (\beta_3 + \beta_2)/2}$, which suggests that, whether or not there exists a scoring function in $S_K^\varepsilon$, which treats the groups $C_2, C_3$ of the same rank, depends upon two things, the size of the groups and also their position on the ROC curve. Specifically, if $\beta_3 - \beta_2 \geq \frac{2\varepsilon p(1 - (\beta_3 + \beta_2)/2)}{\lambda(C_3)}$, then $\tilde{s} \notin S_K^\varepsilon$. The following lemma formalises this intuition, the proof follows via a direct generalisation of the above example.

**Lemma 2.2.** *Let $\Delta > 0$, $i \in [K]$ and define $S_K^{(i,\Delta)}$ as the set of scoring functions such that, for all $s \in S_K^{(i,\Delta)}$, one has that $\forall j : |\mu_j - \mu_i| \geq \Delta$, $\mathrm{sign}(s(i) - s(j)) = \mathrm{sign}(\mu_i - \mu_j)$. There exist a $\tilde{s} \in S_K^{(i,\Delta)}$, $\nu \in \mathcal{B}$, such that on problem $\nu$, $d_\infty(\eta, \tilde{s}) \geq \frac{\Delta|\{j : |\mu_i - \mu_j| \leq \Delta\}|}{p(1 - \mu_i)}$.*

Lemma 2.2 suggests that the for $i \in [K]$ the learner must be *at least* able to distinguish $\mathcal{H}_0^{i,j}$ vs $\mathcal{H}_1^{i,j}$, for all $j : |\mu_i - \mu_k| \leq \tilde{\Delta}_i$, where, $\tilde{\Delta}_i := \max \left\{ x > 0 : \sum_{i \neq j} x\mathbb{I}(|\mu_i - \mu_j| \leq x) < K\varepsilon p(1 - \mu_i) \right\}$.

The following Lemma shows that $\tilde{\Delta}_i$ is not only an upper bound on the necessary order of the confidence level around $\mu_i$ but also a lower bound, the proof of which can be found in the proof of Theorem 3.2.

**Lemma 2.3.** *For a problem $\nu \in \mathcal{B}$, let $s \in S_K$ be a scoring function such that the following holds, $\forall i \in [K]$,*

$$\forall j : |\mu_j - \mu_i| \geq \tilde{\Delta}_i/4, \; \mathrm{sign}(s(i) - s(j)) = \mathrm{sign}(\mu_i - \mu_j)$$

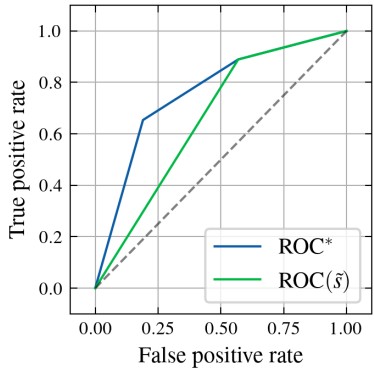

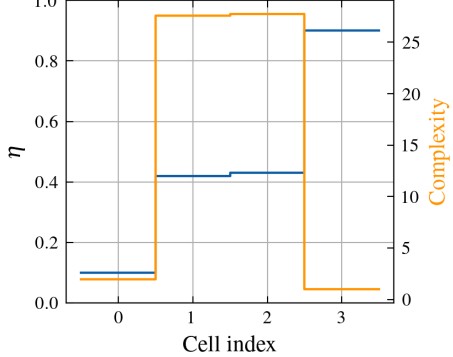

Figure 1: The ROC curves of $\eta$ and $\tilde{s}$, as defined in Equations (2), with $(\beta_1, \beta_2, \beta_3) = (0.1, 0.2, 0.6)$ and $C_1, C_2, C_3$ equally sized.

Figure 2: The complexity of points $i \in [K]$, for $\eta$ as defined in Scenario 1 of the experiments, see Section 4.

*then $s \in S_K^\varepsilon$.*

Under the conditions where $K\varepsilon p > 1$ and for a grid point $i \in [K]$, $\mu_i$ is close to one, a certain phenomenon occurs. Specifically, when $\tilde{\Delta}_i \geq 1 - \mu_i$, for the learner to know the value of $\mu_i$ up to an error of $\tilde{\Delta}_i$ is no longer sufficient, as in such a case, to the best of the learners knowledge, the value of $\mu_i$ may be arbitrarily close to 1, and thus have an arbitrarily large effect on the regret. With this in mind, we define $\Delta_i$ as follows,

$$\Delta_i := \max\left\{x > 0 : \sum_{i \neq j} x\mathbb{I}\big(|\mu_i - \mu_j| \leq x\big) < K\varepsilon p(1 - \mu_i)\right\} \wedge (1 - \mu_i)$$

Note that in the case where $K\varepsilon p < 1$, we have $\Delta_i = \tilde{\Delta}_i$. In the case where $\mu_i \geq \Delta_i$, we thus define the complexity at a point $i \in [K]$ as $H_i^{(1)} = \frac{1}{\mathrm{kl}(\mu_i, \mu_i + \Delta_i) \wedge \mathrm{kl}(\mu_i, \mu_i - \Delta_i)}$. If $\mu_i \leq \Delta_i$, $H_i^{(1)} = \frac{1}{\mathrm{kl}(\mu_i, \mu_i + \Delta_i)}$. For a problem $\nu \in \mathcal{B}$, the total problem complexity is then given as $\sum_{i \in [K]} H_i^{(1)}$. As an illustrative example, see Figure 2 for the complexity of $i \in [K]$ with $\eta$ defined as in Scenario 1 of the experiments, see section 4.

There is a natural comparison to multi armed bandits, where the problem complexity is typically given as the summation across the individual complexity of each arm. However, in most multi armed settings the complexity of a single arm is dependent upon its distance to a single other arm, e.g. the optimal arm, whereas in our setting the complexity of a single grid point $i \in [K]$ has a more complex dependency on the shape of the posterior around $G_i$.

## 2.3 Performance of the passive approach

At this point we can consider how a uniform sampling strategy would perform, that is, the learner simply draws a sample from each section of the grid in turn. This would be essentially a *passive approach*, in line with the classical batch setting. For a uniform/passive sampling strategy to be PAC$(\varepsilon, \delta)$ one would have to draw samples until the width of the confidence interval at all points $i$ is less than $\Delta_i$. Therefore, a uniform sampling strategy, with an appropriate stopping rule, would have the following tight upper bound on its expected sampling time, up to log terms, $cK \max_{i \in [K]} \frac{1}{\mathrm{kl}(\mu_i, \mu_i + c'\Delta_i)}$, for some absolute constants $c, c' > 0$. The improvement we hope to gain in the active setting is to replace the max with a weighted summation across the grid. Thus, in settings where the $\Delta_i$ are relatively constant across large sections of the grid, then the theoretical performance of a passive approach can be close to optimal. In contrast, in cases where a very small section of the interval is hard to rank and the rest is easy - i.e. $K \max_{i \in [K]} \frac{1}{\mathrm{kl}(\mu_i, \mu_i + \Delta_i)}$ is much greater than $\sum_{i \in [K]} \frac{1}{\mathrm{kl}(\mu_i, \mu_i + \Delta_i)}$, a passive approach will fail. Such settings involve large $K$ where the gaps $\Delta_i$ on the majority of cells are large, with a relatively small number of cells with small gaps $\Delta_i$. Incidentally, we point out that this corresponds to many situations of interest in practice (in

information retrieval, for a specific request, the vast majority of the documents are equally irrelevant, while the ranking of a very small fraction of relevant documents is challenging; the same phenomenon is also observed in credit-risk screening). In these cases the benefit to the practitioner, will be that they quickly focus on the interesting sections of the feature space.

## 2.4 Related literature in active learning

While, to the best of our knowledge, Bipartite Ranking has not yet been considered under active learning, there are several related settings. Firstly, it is important to note, that as we assume the posterior $\eta$ is piecewise constant on the grid of size $K$, we can view our problems as a $K$ armed bandit. In the case where $K = 2$ the bipartite ranking problem becomes akin to best arm identification (BAI) for the two armed bandit, also known as A/B-Testing. In BAI for the two armed bandit, the learner sequentially draws samples from two distributions $\nu_A, \nu_B$ with respective means $\mu_A, \mu_B$. Their objective is then carry out the hypothesis test, $\mathcal{H}_0 := (\mu_A \leq \mu_B)$, $\mathcal{H}_1 := (\mu_a > \mu_B)$ in as few samples as possible. A/B Testing is considered in an active, fix confidence regime, in Kaufmann and Kalyanakrishnan (2013), wherein they prove a lower bound on the expected sampling time of any PAC$(\varepsilon, \delta)$ as of the order $\frac{\log(1/\delta)}{\mathrm{kl}(\mu_A, \mu_B)}$. Note that, in the case where $K = 2$ `active-rank` matches said lower bound up to logarithmic terms. In the fixed confidence regime, the BAI problem has also been generalised for larger $K > 2$ - see Garivier and Kaufmann (2016),Jamieson and Talwalkar (2016) along with the TopM problem, where the learner must output the $M$ best arms - see Kaufmann et al. (2014), Kalyanakrishnan et al. (2012), however, for $K > 2$ both BAI and TopM problems are no longer comparable to our setting.

Aside from BAI and the TopM problem, there are several other settings in active learning that, while not directly comparable to our own, are worth mentioning. The first is active clustering. Several works have considered clustering in an online framework, see Choromanska and Monteleoni (2012), Liberty et al. (2016), Cohen-Addad et al. (2021) and Khaleghi et al. (2012). In the above works new observations from certain arms become available to the learner at each time step, however the learner does not actively choose which arms to pull and therefore the flavour of the above literature is very different to our setting. Much closer is the work of Yang et al. (2022) in which the authors consider a active clustering problem, represented as $K$ armed $d$ dimensional bandit, where the arms are split into $M$ clusters. They work in a PAC$(\delta)$ setup where their goal is to recover the entire clustering of the arms, with probability greater than $1 - \delta$ in as few samples as possible. Comparing to our setting, if one is to view a section of the grid on which $\eta$ is constant as a single cluster, by retrieving the clustering of the arms one can then easily do ranking. Their results differ to our own in several key ways though. Firstly their algorithm takes the number of clusters $M$ as a parameter, this highlights the main difference between their setting and our own. In the Bipartite ranking problem, assuming $\varepsilon$ is not very small, one does not have to recover exactly all the clusters to ensure regret under the $d_\infty$ norm is less than $\varepsilon$. Therefore we do not need to know the number of clusters and our algorithm must be able to exploit larger $\varepsilon$ to achieve smaller stopping times. The second key difference is that the results of Yang et al. (2022) are hold only in the asymptotics, that is as $\delta \to 0$. Their algorithm employs a forced exploration phase, which ensures each arm is pulled at at least a sub linear rate. Essentially, this means that in such an asymptotic setting, *the means of the arms are known to the learner*, which naturally drastically changes the nature of their results. Extension to bounds for fixed $\delta > 0$ would be none trivial, noted as potential future work in Yang et al. (2022), and essential if one were to compare to our confidence setting.

Also of note is active multi class classification. In Krishnamurthy et al. (2017) they consider a cost sensitive classification problem, where the learner receives input examples $x \in \mathcal{X}$ and cost vectors $c \in \mathbb{R}^K$, where $c(y)$ is the cost of predicting label $y$ on $x$. For each input example received the learner is able to query a subset of labels. The objective is to then train a classifier with minimal expected loss in as few queries as possible. The results of Krishnamurthy et al. (2017) cannot be directly compared to our own, as in our setting there is no such thing as the cost of a classifier at a given point $x \in \mathcal{X}$, as the cost miss ranking a section of $[0, 1]$ is dependent on our ranking of the entire feature space.

## 3 Our results

### 3.1 The `active-rank` Algorithm

Our algorithm `active-rank` maintains an active set of grid points across several rounds. At the beginning of each round `active-rank` draws a sample, uniformly, from all points of the grid in the active set and at the end of each round, eliminates points from the active set based on a specific criterion. We track the empirical mean of all samples drawn from the $i$th point of the grid, $[G_i, G_{i+1})$ up to round $t$ as $\hat{\mu}_i^t$. At the beginning of each round $t$, for each grid point $i \in [K]$, we will maintain an upper and lower confidence bound, on $\mu_i$, which we term the UCB and LCB index respectively. At time $t$, for each grid point $i \in [K]$ and exploration parameter $\beta(t, \delta) : \mathbb{N} \times [0, 1] \to \mathbb{R}_+$, remaining in the active set, we then define the LCB index,

$$\mathrm{LCB}(t, i) := \min \left\{ q \in [0, \hat{\mu}_i^t] : \mathrm{kl}(\hat{\mu}_i^t, q) \leq \frac{\beta(t, \delta)}{t} \right\}, \tag{3}$$

and the UCB index,

$$\mathrm{UCB}(t, i) := \max \left\{ q \in [\hat{\mu}_i^t, 1] : \mathrm{kl}(\hat{\mu}_i^t, q) \leq \frac{\beta(t, \delta)}{t} \right\}. \tag{4}$$

Let $S_t$ denote the active set at the beginning of round $t$, via careful choice of exploration parameter the following Lemma holds, the proof of which can be found in Section A of the supplementary material.

**Lemma 3.1.** *We have that, the event*

$$\mathcal{E} = \bigcap_{t \in \mathbb{N}} \bigcap_{i \in [S_t]} \{\mu_k \in [\mathrm{LCB}(t, i), \mathrm{UCB}(t, i)]\},$$

*occurs with probability greater than* $1 - \delta$.

For $i \in [K]$, time $t$ and $z \in \mathbb{R}$ define, $U_{i,t}(z) := \{j \in S_t : j \neq i, |\hat{\mu}_i^t - \hat{\mu}_j^t| \leq z\}$. Following Lemma 2.2, our intuition would be to then remove a point $i$ from the active set if, $|U_{i,t}(\mathrm{UCB}(i, t) - \mathrm{LCB}(i, t))| \leq \frac{cK\varepsilon p(1 - \hat{\mu}_{i,t})}{|\mathrm{UCB}(i,t) - \mathrm{LCB}(i,t)|}$, for some well chosen constant $c > 0$. However, due to the technical difficulty of the proof, we make the following concession. For $t > 0$, let $S_t$ be the list $S$ at time $t$. At time $t$ let $\Delta_{(t)} = \max_{i \in S_t}(\mathrm{UCB}(t, i) - \mathrm{LCB}(t, i))$. Furthermore, at time $t$ define the set of grid points,

$$\mathcal{Q}_t := \left\{ i \in [K] : \Delta_{(t)} \leq \frac{1}{24} \left( \frac{K\varepsilon \hat{p}_t}{|U_{i,t}(6\Delta_{(t)})|} \wedge 1 \right) (1 - \hat{\mu}_i^t) \right\}. \tag{5}$$

If a point exists in $\mathcal{Q}_t$, we remove it from the active set. Note that `active-rank` does not take the average of the posterior $p$ as a parameter. Instead we show it is possible to use an estimate $\hat{p}_t$ which updates round by round.

Elimination algorithms such as `active-rank` have seen wide usage in the literature for BAI, see In Paulson (1964), Mannor and Tsitsiklis (2004), Even-Dar et al. (2002) and Even-Dar et al. (2006). However, closer to our work is the Racing algorithm Kaufmann and Kalyanakrishnan (2013), designed for the TopM problem, where, as in our approach, the confidence bounds used are based on the kl divergence as opposed to Hoeffdings. Their elimination criterion, however, differs considerably to our own. For simplicity let us consider the Top1 problem, that is BAI - the following arguments can be extended in the case of TopM. The racing algorithm of Kaufmann and Kalyanakrishnan (2013) eliminates an arm $i \in [K]$ from the active set, at time $t$, when, the positive gap the lower confidence bound around the highest empirical mean and the upper confidence bound at point $i$ is greater than $\varepsilon$. However, due to the global nature of the ranking problem, in our setting, the decision to remove a point from the active set is not made based on the distance to another single point. We rather consider a condition on the local smoothness of the posterior around the point $i$. An additional difficulty that arises here is that the local smoothness around a point can potentially depend upon points no longer in the active set and once a point is no longer in the active set, we essentially have no control of the width of its confidence interval.

### 3.2 Theoretical bounds

**Algorithm 1** `active-rank`

---
**Initialise:** $S = [K], t = 1$
**repeat**
  **for** $i \in S$ **do**
    Sample a point drawn uniformly from $[G_i, G_{i+1})$
  **end for**
  Let $p_t$ be a point drawn uniformly from $[0, 1]$ and update $\hat{p}_t = ((t-1)\hat{p}_{t-1} + p_t)/t$
  **for** $i \in S$ **do**
    **if** $i \in \mathcal{Q}_t, \Delta_{(t)} \leq \hat{p}_t/4$ **then**
      $S = S \setminus \{i\}$
    **end if**
  **end for**
  $t = t + 1$
**until** $S = \emptyset$
Let $\hat{\sigma}_t \in \mathfrak{G}_K$ be the permutation sorting the list $(\widehat{\mu}_i^t)_{i \in [K]}$ into ascending order.
**Output:** $\hat{s} = \sum_{i \in K} i \cdot \mathbb{I}\{x \in [G_{\hat{\sigma}(i)}, G_{\hat{\sigma}(i+1)})\}$

---

**Proving** `active-rank` **is PAC**$(\varepsilon, \delta)$ **and upper bounding the expected sampling time** Theorem 3.2 demonstrates that our algorithm `active-rank` is PAC$(\varepsilon, \delta)$ and provides a problem dependent upper bound on it's expected sampling time. The proof can be found in Section A of the supplementary material. Theorem 3.2 makes no assumption on the posterior $\eta$, aside from it being piecewise constant on the grid of size $K$, i.e. Assumption 2.1. For $i \in [K]$, set $H_i^{(2)} = \max_{j \in [K]} \left( \frac{1}{\mathrm{kl}(\mu_j, \mu_j + \Delta_i/8)} \vee \frac{1}{\mathrm{kl}(\mu_j, \mu_j - \Delta_i/8)} \right)$.

**Theorem 3.2.** *For $\varepsilon, \delta > 0, \gamma > 480/\log(K)$, with $\beta(t, \delta) = c_\gamma \log(t^2 K^2/\delta)$ where $c_\gamma$ is a constant depending only on $\gamma$, on all problems $\nu \in \mathcal{B}$, on execution of* `active-rank`*, with output $\hat{s}$, we have that,*

$$d_\infty(\hat{s}, \eta) \leq \varepsilon ,$$

*with probability greater than $1 - \delta$. Furthermore, the expected stopping time of* `active-rank` *is upper bounded by the following,*

$$c'_\gamma \sum_{i \in [K]} H_i^{(2)} \log\left( c''_\gamma H_i^{(2)} K/\delta \right) ,$$

*where $c'_\gamma$, $c''_\gamma$, are constants depending only on $\gamma$.*

**Lower bound** Theorem 3.3 provides a problem dependent lower bound on the expected sampling time of any PAC$(\varepsilon, \delta)$ policy. The proof of Theorem 3.3 can be found in Section B of the supplementary material.

**Theorem 3.3.** *Let $\varepsilon \in [0, 1/4), 0 < \delta < 1 - \exp(-1/8)$ and $\nu \in \mathcal{B}$ such that $K\varepsilon p < 1/8$. For any PAC$(\varepsilon, \delta)$ policy $\pi$, there exists a problem $\bar{\nu} \in \mathcal{B}$ such that, for all $i \in [K]$, $\bar{\Delta}_i \geq \Delta_i/2$ where $\bar{\Delta}_i$ is the gap of the $i$th grid point on problem $\bar{\nu}$, where the expected stopping time of policy $\pi$ on problem $\bar{\nu}$ is bounded as follows,*

$$\mathbb{E}_{\bar{\nu}, \pi}\left[\tau_\pi^{\bar{\nu}}\right] \geq c' \sum_{i \in [K]} \bar{H}_i^{(1)} ,$$

*where $c' > 0$ is an absolute constant and $\bar{H}_i^{(1)}$ is the complexity of point $i$ on problem $\bar{\nu}$. Furthermore we have that,*

$$\sum_{i \in [K]} \bar{H}_i^{(1)} \geq c \sum_{i \in [K]} H_i^{(1)} ,$$

*where $c > 0$ is an absolute constant.*

The proof of Theorem 3.3 follows from a novel application of a Fano type inequality on a well chosen set of problems.

**Gap between upper and lower bound** There are essentially two components in the gap between the bounds of Theorems, 3.3 and 3.2. The first is the additional logarithmic dependency upon $K$ present in our upper bound. Despite being logarithmic this dependence is potentially significant, as in practical situations, the size of grid needed, for the assumption that the posterior $\eta$ is piecewise constant, may be very large. The second component, in the gap between upper and lower bounds is the difference in the $H_i^{(1)}$ and $H_i^{(2)}$ terms. The reason $H_i^{(2)}$ appears in Theorem 3.2 is that, the decision to remove a point $i \in [K]$ from our active set is made based on the minimum width of confidence interval across the entire grid, $\Delta_{(t)}$ as opposed to the local width at $i$, $\mathrm{UCB}(i, t) - \mathrm{LCB}(i, t)$, see Equation (5). As we are dealing with Bernoulli distributions and kl divergence based confidence bounds, for a fixed number of samples, points close to zero or one will have tighter confidence bounds

and thus may be sampled more than is necessary. If one were to assume that the posterior $\eta$ exists solely in the interval $[\gamma, 1 - \gamma]$ for some $\gamma > 0$, then for all $i \in [K]$, $H_i^{(2)}$ and $H_i^{(1)}$ will be with a constant factor of each other, with that constant depending on $\gamma$.

In authors opinion, both the logarithmic dependency on $K$ and usage of $\Delta_{(t)}$ may be removed. However, this would require several non trivial modifications to the proof of Theorem 3.2, see the discussion in Section A of the supplementary material for details. As, to the best of our knowledge, this is the first work to consider the bipartite ranking problem in an active learning setting, we present Theorem 3.2 as it stands and leave the aforementioned improvements for future works.

## 4    Experiments

In this section we discuss practical cases based on synthetic data. For all experiments $\delta$ is fixed at 0.01 and the constants used are smaller than their theoretical counterparts, which are typically overestimated, furthermore $\widehat{p}$ is calculated with all previous samples. As represented in Figure 3, each cell $i$ is assigned a level value $\mu_i$ so that $\eta$ follows the Assumption 2.1. Without loss of generality, we assume that $\eta$ can be described as an increasing family $(\mu_i)_{i \in [K]}$. Our study scenarios are then as follows:

*   Scenario 1: $(\mu_i)_{i \in [K]} = (0, 0.28, 0.3, 0.38)$ and $K = 16$;
*   Scenario 2: $(\mu_i)_{i \in [K]} = (0.8((i-1)/K)^4 + 0.1)_{i \in [K]}$ and $K = 64$;
*   Scenario 3: $(\mu_i)_{i \in [K]}$ is sub-sampled (with replacement) of $((i-1)/K)^4)_{i \in [100]}$ and $K = 64$
*   Scenario 4: $\mu_i = 0.8((i-1)/K) + 0.1 \, \forall i \in [K] \backslash \{7, 8\}$ with $\mu_7 = 0.8(6/K) + 0.3$, $\mu_8 = 0.8(7/K) - 0.1$ and $K = 16$

The objective of these scenarios is to evaluate the capacity of the algorithm on different cases. As shown in Figure 3, scenarios 1 and 3 will have variable jumps and cell sizes.

**Competing algorithms**    To our knowledge there is no algorithm dealing with the active learning for bipartite ranking problem, we thus compare to the following naive approaches. **Passive rank**: each new point $a_t$ is drawn uniformly on $[0, 1]$. **Naive rank**: each new point $a_t$ is sampled in $P_{i_t}$ s.t. $i_t := \arg\max_{i \in [K]} \text{UCB}(t, i) - \text{LCB}(t, i)$, this algorithm reduces the bias in an undifferentiated way without considering the problem as global (requiring peer-to-peer comparison). **Active classification**: for this algorithm we consider binary classification with threshold 0.5. The set of active cells $S$, as defined in Algorithm 1 is then $S_t = \{i \in [K]; 0.5 \in [\hat{\mu}_i^t - \text{LCB}(t, i); \hat{\mu}_i^t + \text{UCB}(t, i)]\} \cup \left\{i; \underset{i \in [K]}{\arg\min}(|0.5 - \hat{\mu}_i^t|)\right\}$. As the competing algorithms do not output a stopping time, for a single $\eta$, algorithm `active-rank` is run across several values of $\varepsilon$, following a geometric sequence of common ratio 0.99 and initial value 1. The values of $\varepsilon$ then plotted against the respective stopping times of `active-rank`.

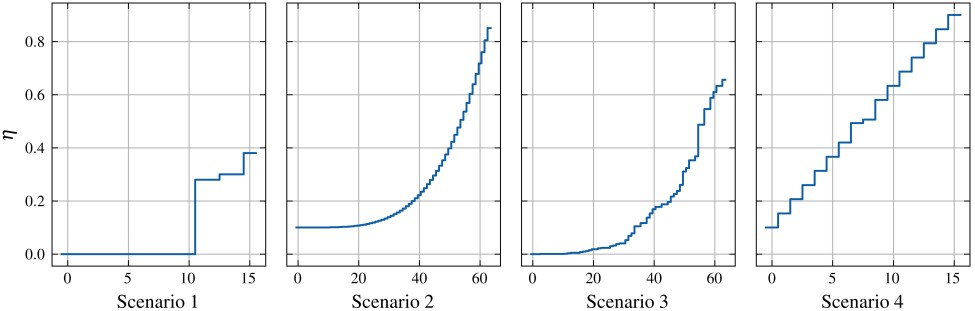

Figure 3: Different scenarios chosen for the experiments

**Interpretation of results**    On scenario 1, the simplest case, and to an extent scenario 2, `active-rank` and passive suffer near identical regret for larger sample sizes. On all other scenarios `active-rank` outperforms all competitors. The fact that `active-rank` does not reach extremely small values of regret suggests that analysis for much higher sample size may be interesting, however this creates issues in computation time, the same goes for larger $K$. Also, the uniform

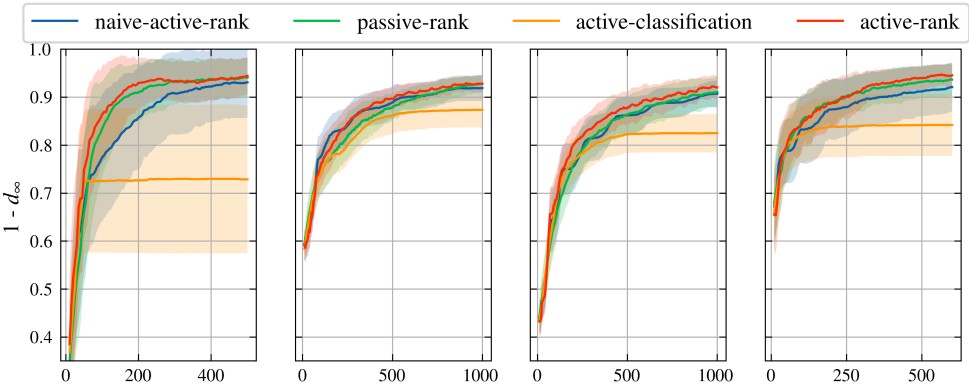

Figure 4: Regret estimated by Monte Carlo for 100 realizations of each algorithm, corresponding to the scenarios of the figure 3 respectively .

approach still performs relatively well, and appears difficult to fool. Some more work may be needed to find a setting in which uniform sampling suffers considerably.

## 5    Conclusion

To the best of our knowledge we have developed the first rigorous framework for the active bipartite ranking problem and our algorithm, `active-rank`, is the first to tackle said problem. Our upper bound on performance of `active-rank` matches our lower bound up to logarithmic terms, in the case where the posterior $\eta$ is not very close to 0 or 1 at any point. As well as theoretical guarantees we have demonstrated good practical performance of `active-rank`, on synthetic data, in various settings. We conclude with some perspectives for future research.

An obvious path for future research, is to replace the Assumption 2.1 with a smoothness assumption on the posterior $\eta$, e.g. a Hölder condition. The setting would then be equivalent to a continuous armed bandit as opposed to a finite armed bandit. Assuming the learner has knowledge of the Hölder coefficient, a standard approach in continuous armed bandits is to first discretise and then apply classic techniques from finite armed bandits, carefully choosing the discretisation level to balance the discretisation error and classical regret. It is of our opinion that such an approach would not be sufficient in our case. We conjecture that to achieve optimal or near optimal performance the learner must vary the level of discretisation across the feature space, based on the flatness of the posterior function $\eta$ and placement on the ROC curve. Also, under such a Hölder condition, the extension to higher dimensions would no longer be immediate.

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

# A Proof of theorem 3.2

Before the proof of Theorem 3.2 we prove some initial lemmas, the first of which is Lemma 3.1, which lower bounds the probability of our favourable event $\mathcal{E}$ by $1 - \delta$. We prove a slightly extended version of Lemma 3.1. At time $t$, define the LCB index of $\widehat{p}$ as,

$$\text{LCB}(t,0) := \min\left\{ q \in [0, \widehat{p}_t] : \text{kl}(\widehat{p}_t, q) \leq \frac{\beta(t, \delta)}{t} \right\}, \tag{6}$$

and the UCB index,

$$\text{UCB}(t,0) := \max\left\{ q \in [\widehat{p}_t, 1] : \text{kl}(\widehat{p}_t, q) \leq \frac{\beta(t, \delta)}{t} \right\}. \tag{7}$$

Note that as $\widehat{p}_t \in [\min(\mu_i^t), \max(\mu_i^t)]$, we have that,

$$\text{UCB}(0, t) - \text{LCB}(0, t) \leq \Delta_{(t)}.$$

The extended version of Lemma 3.1 is then as follows,

**Lemma A.1.** *We have that, the event*

$$\mathcal{E} = \bigcap_{t \in \mathbb{N}} \bigcap_{i \in [S_t] \cup \{0\}} \left\{ \mu_k \in [\text{LCB}(t, i), \text{UCB}(t, i)] \right\},$$

*occurs with probability greater than $1 - \delta$.*

*Proof.* Via Chernoff's inequality, for $i \in [K] \cup \{0\}$, at time $t$ we have that,

$$\mathbb{P}(\text{LCB}(t, k) \geq \mu_k) \leq \exp(-\beta(t, \delta)),$$

and,

$$\mathbb{P}(\text{UCB}(t, k) \leq \mu_k) \leq \exp(-\beta(t, \delta)).$$

It then remains to note that via our choice of exploration parameter and a union bound,

$$2 \sum_{t \in \mathbb{N}} \sum_{k \in [K] \cup \{0\}} \exp(-\beta(t, \delta)) \leq \delta,$$

and the result follows. $\square$

The following Lemma shows that, on rounds in which a point is removed from the active set, our estimate $\widehat{p}$ remains within a constant factor of the true $p$.

**Proposition A.2.** *On event $\mathcal{E}$ we have that, for all rounds $t$ such that a point is removed from the active set, $2p/3 \leq \hat{p}_t \leq 4p/3$.*

*Proof.* Let $t$ be a round in which a point is removed from the active set. On event $\mathcal{E}$, for all $i \in [K]$,

$$|\mu_i - \hat{\mu}_i^t| \leq \Delta_{(t)},$$

we thus have $|p - \hat{p}| \leq \Delta_{(t)}$ which, in combination with the fact $\Delta_{(t)} \leq \hat{p}/4$, implies $\Delta_{(t)} \leq (\widehat{p} + \Delta_{(t)})/3$ and thus $\Delta_{(t)} \leq p/3$.

$\square$

For the purposes of the proof we split Theorem 3.2 into two parts. The first part is covered by the following lemma, which states that `active-rank` is PAC($\varepsilon, \delta$).

**Lemma A.3.** *For $\varepsilon, \delta > 0$, $1/K \geq \varepsilon p$ on all problems $\nu \in \mathcal{B}$, on execution of `active-rank`, with output $\hat{s}$, we have that,*

$$d_\infty(\hat{s}, \eta) \leq \varepsilon,$$

*with probability greater than $1 - \delta$.*

*Proof.* For the remainder of this proof we will work under the following event,

$$\mathcal{E} = \bigcap_{t \in \mathbb{N}} \bigcap_{k \in [K]} \{\mu_k \in [\text{LCB}(t, i), \text{UCB}(t, i)]\} .$$

**Proposition A.4.** *Let $i \in [K]$ be removed from $S_t$ at time $t$, such that $\{j : |\mu_i - \mu_j| \le \Delta_i\} \subset S_t$. For all $j : |\mu_i - \mu_j| \le \Delta_j$, we have that, $\forall k \in S_t : |\mu_k - \mu_j| \ge \Delta_j$,*

$$\text{sign}(\widehat{\mu}_j^\tau - \widehat{\mu}_k^\tau) = \text{sign}(\mu_j - \mu_k) .$$

*Proof.* Firstly note that, for all $s \ge t$, $\Delta_{(s)} \le \Delta_{(t)}$. Thus for all $j \in S_t$, $|\widehat{\mu}_j^\tau - \mu_j| \le \Delta_{(t)}$ and therefore for any point $j : 4\Delta_{(t)} \le \Delta_j$, we have that, for all $k \in S_t : |\mu_j - \mu_k| \le \Delta_j$,

$$\text{sign}(\widehat{\mu}_j^\tau - \widehat{\mu}_k^\tau) = \text{sign}(\mu_j - \mu_k) .$$

Thus to prove our result we must show that, for all $j : |\mu_i - \mu_j| \le \Delta_j$, $\Delta_j \le 4\Delta_{(t)}$. Now, if $i$ is removed from $S_t$ at time $t$, we have that,

$$\Delta_{(t)} \le \frac{1}{24} \left( \frac{K\varepsilon \widehat{p}_t}{|U_{i,t}(6\Delta_{(t)})|} \wedge 1 \right) (1 - \widehat{\mu}_i^t) ,$$

thus,

$$\Delta_{(t)} \le \frac{1}{24}(1 - \widehat{\mu}_i^t) , \tag{8}$$

and therefore via Proposition A.2,

$$|U_{t,i}(6\Delta_{(t)})| \le \frac{K\varepsilon \widehat{p}_t (1 - \widehat{\mu}_i^t)}{24\Delta_{(t)}} \le \frac{K\varepsilon p(1 - \widehat{\mu}_i^t)}{12\Delta_{(t)}} \le \frac{K\varepsilon p(1 - \mu_i)}{6\Delta_{(t)}} ,$$

and we have that $6\Delta_{(t)} \le \Delta_i$. Now let $j : |\mu_j - \mu_i| \le 4\Delta_{(t)}$, we then have, $|\widehat{\mu}_j - \widehat{\mu}_i| \le 2\Delta_{(t)}$ and thus $U_{j,t}(4\Delta_{(t)}) \subset U_{i,t}(6\Delta_{(t)})$ and furthermore $\Delta_{(t)} \le \frac{1}{12}(1 - \widehat{\mu}_j^t)$ therefore, via proposition A.2,

$$|U_{t,j}(4\Delta_{(t)})| \le \frac{K\varepsilon \widehat{p}_t (1 - \widehat{\mu}_i^t)}{24\Delta_{(t)}} \le \frac{K\varepsilon p(1 - \widehat{\mu}_i^t)}{12\Delta_{(t)}} \le \frac{K\varepsilon p(1 - \mu_j)}{6\Delta_{(t)}} ,$$

and as a result $\Delta_j \le 4\Delta_{(t)}$. Now take $j : |\mu_j - \mu_i| \ge 4\Delta_{(t)}$. If $\Delta_j \le 4\Delta_{(t)}$ we are done. Thus assume $\Delta_j \ge 4\Delta_{(t)}$. In this case, we have that $j : |\mu_i - \mu_j| \ge \Delta_j$. Thus we have shown that, for a point $j \in S_t$, if $|\mu_i - \mu_j| \le \Delta_j$, then $\Delta_{(t)} \le \Delta_j/4$ and the result follows. □

**Proposition A.5.** *At time $t$, let $i \in S_t$, we have that, $\forall j \in [K] \setminus S_t : |\mu_i - \mu_j| \ge \Delta_i$,*

$$\text{sign}(\widehat{\mu}_j^\tau - \widehat{\mu}_i^\tau) = \text{sign}(\mu_j - \mu_i) .$$

*and furthermore, $\forall j : \{k : |\mu_j - \mu_k|\} \not\subset S_t, \forall k \in [K] : |\mu_k - \mu_j| \le \Delta_j$,*

$$\text{sign}(\widehat{\mu}_j^\tau - \widehat{\mu}_k^\tau) = \text{sign}(\mu_j - \mu_k) .$$

*Proof.* The proof will follow by induction. Assume at time $t$ that $\forall i \in S_{t-1}$ we have that, $\forall j \in [K] \setminus S_{t-1} : |\mu_i - \mu_j| \ge \Delta_i$,

$$\text{sign}(\widehat{\mu}_j^\tau - \widehat{\mu}_i^\tau) = \text{sign}(\mu_j - \mu_i) ,$$

and that, $\forall j \in [K] : \{k : |\mu_j - \mu_k|\} \not\subset S_{t-1}, \forall k : |\mu_k - \mu_j| \le \Delta_j$,

$$\text{sign}(\widehat{\mu}_j^\tau - \widehat{\mu}_k^\tau) = \text{sign}(\mu_j - \mu_k) .$$

Now let $\tilde{i} \in S_t$, and $j \in [K] \setminus S_t : |\mu_{\tilde{i}} - \mu_j| \ge \Delta_{\tilde{i}}$, we must show that,

$$\text{sign}(\widehat{\mu}_j^\tau - \widehat{\mu}_{\tilde{i}}^\tau) = \text{sign}(\mu_j - \mu_{\tilde{i}}) ,$$

and that, $\forall j \in [K] : \{k : |\mu_j - \mu_k|\} \not\subset S_t, \forall i : |\mu_k - \mu_j| \le \Delta_j$,

$$\text{sign}(\widehat{\mu}_j^\tau - \widehat{\mu}_k^\tau) = \text{sign}(\mu_j - \mu_i) .$$

For the first statement, assume $j \in S_{t-1}$ and that $\{k : |\mu_j - \mu_k|\} \subset S_{t-1}$, as otherwise we are done, via the inductive assumption. The point $j$ is then removed from the active set at the end of round $t + 1$ and the statement thus follows via Proposition A.4.

For the second statement, for $j \in S_{t-1}$ assume $\{k : |\mu_j - \mu_k|\} \subset S_{t-1}$, as otherwise we are done via the inductive assumption. Therefore, there must exist such an $k$, removed from the active set at the end of round $t - 1$, such that $|\mu_k - \mu_j| \leq \Delta_j$. The result then follows via Proposition A.4. $\quad\square$

For a set $C \subset [0, 1]$, define,

$$\kappa(C) := \frac{1}{\lambda(C)} \int_C \eta(x)\, , dx\, .$$

Let $\alpha \in [0, 1]$, define the subset $Z_\alpha \subset [0, 1]$ such that, $\mathbb{P}(X \in Z_\alpha | Y = -1) = \alpha$, that is,

$$\frac{\lambda(Z_\alpha)(1 - \kappa(Z_\alpha))}{1 - p} = \alpha\, ,$$

and such that, for some $i_\alpha \in [K]$,

$$\forall j : \mu_j > \mu_{i_\alpha}, P_j \subset Z_\alpha\, , \qquad \forall j : \mu_j < \mu_{i_\alpha}, P_j \cap Z_\alpha = \emptyset\, . \tag{9}$$

We then have $\mathrm{ROC}^*(\alpha) = \mathbb{P}(X \in Z_\alpha | Y = +1) = \frac{\lambda(Z_\alpha)(\kappa(Z_\alpha))}{p}$. The choice of $Z_\alpha$ is not necessarily unique, and as $\eta$ me be constant across multiple sections of the grid $i_\alpha$ is also not necessarily unique, in this case we take arbitrary $Z_\alpha, i_\alpha$. Now define the subset $\hat{Z}_\alpha \in [0, 1]$ such that,

$$\frac{\lambda(\hat{Z}_\alpha)(1 - \kappa(\hat{Z}_\alpha))}{1 - p} = \alpha\, ,$$

and,

$$\forall x \in \hat{Z}_\alpha, y \notin \hat{Z}_\alpha, s_{\hat{\mathcal{P}}}(x) \geq s_{\hat{\mathcal{P}}}(y)\, ,$$

so $\mathrm{ROC}(s, \alpha) = \frac{\lambda(\hat{Z}_\alpha)(\kappa(\hat{Z}_\alpha))}{p}$. Again $\hat{Z}_\alpha$ is not necessarily unique, in which case we choose arbitrarily. Via Propositions A.4 and A.5, we have that, $\forall j \in [K] : |\mu_j - \mu_{i_\alpha}| \leq \Delta_{i_\alpha}$,

$$\mathrm{sign}(\hat{\mu}_j^\tau - \hat{\mu}_{i_\alpha}^\tau) = \mathrm{sign}(\mu_j - \mu_{i_\alpha})\, . \tag{10}$$

Let

$$Z_\alpha' = \{x \in Z_\alpha : |\eta(x) - \mu_{i_\alpha}| \leq \Delta_{i_\alpha}\}\, , \qquad \hat{Z}_\alpha' = \{x \in \hat{Z}_\alpha : |\eta(x) - \mu_{i_\alpha}| \leq \Delta_{i_\alpha}\}\, .$$

Via Equation 10, we have that,

$$\mathrm{ROC}(\alpha, \eta) - \mathrm{ROC}(\alpha, s_{\hat{\mathcal{P}}}) = \frac{\lambda(Z_\alpha')\kappa(Z_\alpha')}{p} - \frac{\lambda(\hat{Z}_\alpha')\kappa(\hat{Z}_\alpha')}{p}\, .$$

Before finalising the proof we must lower bound $\lambda(\hat{Z}_\alpha')$ and $\kappa(\hat{Z}_\alpha')$. We first lower bound $\kappa(\hat{Z}_\alpha')$.

$$\kappa(\hat{Z}_\alpha') \geq \mu_{i_\alpha} - \Delta_{i_\alpha}\, , \qquad \kappa(Z_\alpha'), \leq \mu_{i_\alpha} + \Delta_{i_\alpha}\, . \tag{11}$$

We will now lower bound $\lambda(\hat{Z}_\alpha')$

$$\frac{\lambda(\hat{Z}_\alpha')}{\lambda(Z_\alpha')} = \frac{1 - \kappa(Z_\alpha')}{1 - \kappa(\hat{Z}_\alpha')} \leq \frac{1 - \mu_{i_\alpha} + \Delta_{i_\alpha}}{1 - \mu_{i_\alpha} - \Delta_{i_\alpha}}\, . \tag{12}$$

Via combinations of Equations (11) and (12), we have,

$$\frac{\lambda(Z_\alpha')\kappa(Z_\alpha')}{p} - \frac{\lambda(\hat{Z}_\alpha')\kappa(\hat{Z}_\alpha')}{p} \leq \frac{1}{p}\Big(\lambda(Z_\alpha')(\mu_{i_\alpha} + \Delta_{i_\alpha}) - \lambda(\hat{Z}_\alpha')(\mu_{i_\alpha} - \Delta_{i_\alpha})\Big) \tag{13}$$

$$\leq \frac{1}{p}\Big(\lambda(Z_\alpha')(\mu_{i_\alpha} + \Delta_{i_\alpha}) - \lambda(Z_\alpha')\frac{(\mu_{i_\alpha} - \Delta_{i_\alpha})(1 - \mu_{i_\alpha} + \Delta_{i_\alpha})}{1 - \mu_{i_\alpha} - \Delta_{i_\alpha}}\Big) \tag{14}$$

$$\leq \frac{2\lambda(Z_\alpha')\Delta_{i_\alpha}}{p(1 - \mu_{i_\alpha} - \Delta_{i_\alpha})} \leq \frac{2\lambda(Z_\alpha')\Delta_{i_\alpha}}{p(1 - \mu_{i_\alpha})} \tag{15}$$

It remains to remark that, $\Delta_{i_\alpha} \leq \frac{\varepsilon p}{\lambda(Z'_\alpha)}$, by definition, and thus,

$$\mathrm{ROC}(\alpha, \eta) - \mathrm{ROC}(\alpha, s_{\hat{\mathcal{P}}}) \leq \varepsilon .$$

As we chose $\alpha$ w.l.o.g the proof then follows.

$\square$

## A.1 Proof of stopping time for `active-rank`

We will now prove the second part of Theorem 3.2, the upper bound on the expected sampling time of `active-rank`.

**Lemma A.6.** *For $\varepsilon, \delta > 0, \gamma > 480/\log(K)$, $1/K \geq p\varepsilon$, with $\beta(t, \delta) = c_\gamma \log(t^2 K^2/\delta)$ where $c_\gamma$ is a constant depending only on $\gamma$, on all problems $\nu \in \mathcal{B}$ such that $\forall i \in [K], \Delta_i \leq 1 - \mu_i$, we have that, for $\gamma > 1$, the expected stopping time of `active-rank` is upper bounded by the following,*

$$c'_\gamma \sum_{i \in [K]} H_i^{(2)} \log\left(c''_\gamma H_i^{(2)} K^2/\delta\right) ,$$

*where $c'_\gamma$, $c''_\gamma$ are constants depending only on $\gamma$.*

*Proof.* We upper bound $\mathbb{E}[\tau]$ as follows. We denote the number of times a section of the grid $[G_i, G_{i+1})$ has been sampled by the learner, up to and including time $t$ as, $N_i(t)$. Let $\tau_i = N_i(\tau)$.

$$\mathbb{E}[\tau_i] \leq \sum_{t=1}^{\infty} \mathbb{P}(\tau_i \geq t) \tag{16}$$

$$\leq \sum_{t=1}^{\infty} \mathbb{P}(i \notin \mathcal{Q}_t) . \tag{17}$$

Define the event,

$$\xi_{i,t} := \{\forall j \in S_t, (\mathrm{UCB}(t, j) \leq \mu_j + \Delta_i/96\} \cup \{\forall j \in S_t, \mathrm{LCB}(t, j) \geq \mu_j - \Delta_i/96)\}$$
$$\cup \{p - \Delta_{(t)} \leq \hat{p}_t \leq p + \Delta_{(t)}\} .$$

**Proposition A.7.** *For $i \in [K]$ such that $\Delta_i \leq 2p$, $t > 0$, we have that, on $\xi_{i,t}$, $i \in \mathcal{Q}_t$.*

*Proof.* First note that, under event $\xi_{i,t}$, $\Delta_{(t)} \leq \Delta_i/96$ and thus for a $j \in U_{i,t}(6\Delta_{(t)})$, we have,

$$|\mu_i - \mu_j| \leq \Delta_i/96 + \Delta_i/96 + 6\Delta_i/96 \leq \Delta_i ,$$

thus,

$$|U_{i,t}(6\Delta_{(t)})| \leq |\{j : |\mu_i - \mu_j| \leq \Delta_i\}| \tag{18}$$

and furthermore, $\Delta_{(t)} \leq p/4$ which implies $\hat{p} \leq 2p$. Now, via definition of $\Delta_i$,

$$\Delta_i \leq \left(\frac{Kp\varepsilon}{|\{j : |\mu_i - \mu_j| \leq \Delta_i\}|} \wedge 1\right)(1 - \mu_i) , \tag{19}$$

In the case, $\frac{Kp\varepsilon}{|\{j:|\mu_i-\mu_j|\leq\Delta_i\}|} \geq 1$,

$$\Delta_{(t)} \leq \frac{(1 - \hat{\mu}_i) + \Delta_{(t)}}{96} \leq (1 - \hat{\mu}_i)/48$$

and we have that

$$\frac{Kp\varepsilon}{|\{j : |\mu_i - \mu_j| \geq \Delta_i\}|} \leq (1 + \Delta_{(t)})\frac{K\hat{p}\varepsilon}{|U_{i,t}(6\Delta_{(t)})|} \leq \frac{3K\hat{p}\varepsilon}{2|U_{i,t}(6\Delta_{(t)})|}$$

thus

$$\frac{K\hat{p}\varepsilon}{|U_{i,t}(6\Delta_{(t)})|} \geq \frac{2}{3}$$

finally leading to,

$$\Delta_{(t)} \leq \frac{K\varepsilon\hat{p}_t(1-\widehat{\mu}_i^t)}{24|U_{i,t}(6\Delta_{(t)})|} .$$

Then let us assume $\frac{Kp\varepsilon}{|\{j:|\mu_i-\mu_j|\leq\Delta_i\}|} \leq 1$. In this case,

$$\Delta_{(t)} \leq \Delta_i/96 \leq \frac{K\varepsilon p(1-\mu_i)}{96|U_{i,t}(6\Delta_{(t)})|} \leq \frac{K\varepsilon\widehat{p}((1-\widehat{\mu}_i^t)+\Delta_{(t)})}{48|U_{i,t}(6\Delta_{(t)})|} \leq \frac{K\varepsilon\widehat{p}(1-\widehat{\mu}_i^t)}{24|U_{i,t}(6\Delta_{(t)})|} ,$$

where the third inequality comes from the fact that $\Delta_{(t)} \leq p/4$ which implies $\widehat{p} \leq 2p$.

$\square$

In what follows, assume $i$ is such that $\Delta_i \leq 2p$. Via combination Proposition A.7 and Equation (17), we have that,

$$\mathbb{E}[\tau_i] \leq \sum_{t=1}^{\infty} \mathbb{P}\big(\xi_{i,t}^c\big) .$$

We will now upper bound $\sum_{t=1}^{\infty} \mathbb{P}(\xi_{i,t})$. For $\gamma > 1$ let,

$$T_0^i = \min\left(t : \frac{\beta(t,\delta)}{t} \leq \min_{i\in[K]}\left(\frac{\mathrm{kl}(\mu_i,\mu_i+\Delta_i/96)}{\log(K)\gamma} \wedge \frac{\mathrm{kl}(\mu_i,\mu_i-\Delta_i/96)}{\log(K)\gamma}\right)\right) .$$

We have that, for all $t > T_0^i, j \in S_t$,

$$\mathbb{P}(\mathrm{UCB}(t,j) \geq \mu_j + \Delta_i/96) \leq \mathbb{P}\left(\mathrm{kl}(\hat{\mu}_j^t,\mu_j+\Delta_i/96) \leq \frac{\mathrm{kl}(\mu_j,\mu_j+\Delta_i/96)}{\log(K)\gamma}\right) . \tag{20}$$

Let,

$$r(\gamma) = \{x \in (\mu_j,\mu_j+\Delta_i/96) : \mathrm{kl}(x,\mu_j+\Delta_i/96) = \mathrm{kl}(\mu_j,\mu_j+\Delta_i/96)/(\log(K)\gamma)\} .$$

Consider the function $\phi(x) = \mathrm{kl}(\mu_j+x,\mu_j+\Delta_i/96)$, on the interval $[0,\Delta_i/96]$. Via the properties of the KL divergence, $\phi$ is convex and $\phi(\Delta_i/96) = 0$. As a result, $\phi(x) \leq (1-x)\,\mathrm{kl}(\mu_j,\mu_j+\Delta_i/96)/(2\Delta_i)$, which implies,

$$r(\gamma) \geq \mu_j + \Delta_i\left(\frac{1}{96} - \frac{2}{\log(K)\gamma}\right) \geq \mu_j + \frac{\Delta_i}{192} ,$$

for $\log(K)\gamma > 384$. We now have, for all $t > T_0^i$,

$$
\begin{aligned}
\mathbb{P}\left(\mathrm{kl}(\hat{\mu}_j^t,\mu_j+\Delta_i/96) \leq \frac{\mathrm{kl}(\mu_j,\mu_j+\Delta_i/96)}{\log(K)\gamma}\right) &= \mathbb{P}\big(\mathrm{kl}(\hat{\mu}_j^t,\mu_j+\Delta_i/96) \leq \mathrm{kl}(r(\gamma),\mu_j+\Delta_i/96)\big) \\
&= \mathbb{P}(\hat{\mu}_j^t \geq r(\gamma)) \\
&\leq \exp(-t\,\mathrm{kl}(\mu_j,r(\gamma))) \\
&\leq \exp\left(-\frac{\log(K)\gamma\,\mathrm{kl}(\mu_j,r(\gamma))}{\mathrm{kl}(\mu_j,\mu_j+\Delta_i/96)}\beta(t,\delta)\right) \\
&\leq \exp\left(-\frac{\gamma\,\mathrm{kl}\big(\mu_j,\mu_j+\frac{\Delta_i}{192}\big)}{\mathrm{kl}(\mu_j,\mu_j+\Delta_i/96)}\beta(t,\delta)\right) \\
&\leq \exp(-\log(K)\beta(t,\delta)c_\gamma)
\end{aligned}
$$

where $c_\gamma$ is a constant depending only on $\gamma$. Thus, via Equation (20), for $t \geq T_0^i$

$$\mathbb{P}(\mathrm{UCB}(t,j) \geq \mu_j + \Delta_i/2) \leq \exp(-\log(K)\beta(t,\delta)c_\gamma) ,$$

via similar reasoning we have also that,

$$\mathbb{P}(\mathrm{LCB}(t,j) \leq \mu_j - \Delta_i/2) \leq \exp(-\log(K)\beta(t,\delta)c_\gamma) ,$$

and furthermore, $\mathbb{P}(\widehat{p}_t \notin [p-\Delta_{(t)},p+\Delta_{(t)}]) \leq \exp(-\beta(t,\delta))$, and now via a union bound,

$$\sum_{t=T_0^i+1}^{\infty} \mathbb{P}(\xi_{t,i}^c) \leq c \exp(-c_\gamma) . \tag{21}$$

where $c > 0$ is an absolute constant. Thus we have shown that for a point $i : \Delta_i \leq 2p$,

$$\mathbb{E}[\tau_i] \leq T_0^i + cK \exp(-c_\gamma) .$$

It is then straight forward to note that for any point $i : \Delta_i \geq 2p$,

$$\mathbb{E}[\tau_i] \leq T_0^{i'} + cK \exp(-c_\gamma) ,$$

where $i'$ is any point $i'$ such that $\Delta_{i'} \leq 2p$. It remains to upper bound $T_0^i$. Via definition of the exploration parameter, we have that,

$$T_0^i \leq H_i^{(2)} \log\left(\tilde{c}_\gamma H_i^{(2)} K/\delta\right) ,$$

where $\tilde{c}_\gamma > 0$ is an absolute constant depending only on $\gamma$. Before continuing, we demonstrate the following Lemma-

**Lemma A.8.** *On all problems $\nu \in \mathcal{B}$, we have that,*

$$|\{i : \Delta_i < 2p\}| \geq K/4.$$

*Proof.* Firstly assume that,

$$|\{i : \Delta_i \geq \mu_i\}| \geq K/4$$

and define $i^* = \max\{i : \Delta_i \geq \mu\}$. We then have that,

$$\forall j : \Delta_j \geq \mu_j, |\mu_{i^*} - \mu_j| \leq \Delta_{i^*} ,$$

and thus,

$$\Delta_{i^*} \leq \frac{K\varepsilon p}{|\{j : |\mu_{i^*} - \mu_j| \leq \Delta_{i^*}\}|} \leq \varepsilon p/4 \leq p .$$

Thus we have $|\{i : \Delta_i < p\}| \geq K/4$. Now assume,

$$|\{i : \Delta_i \geq \mu_i\}| \leq K/4 .$$

As we must have $|\{i : \mu_i > 2p\}| \leq K/2$, we thus have, $|\{i : \Delta_i < 2p\}| \geq K/4$ $\qquad\square$

With Lemma A.8, we can then upper bound $\mathbb{E}[\tau]$ as follows,

$$\mathbb{E}[\tau] \leq 4 \sum_{i:\Delta_i \leq 2p} \mathbb{E}[\tau_i] \leq 4 \sum_{i \in [K]} H_i^{(2)} \log(\tilde{c}_\gamma H_i^{(2)} K/\delta) + c_\gamma' K \exp(-c_\gamma) ,$$

and the result follows.

$\qquad\square$

# B    Proof of Theorem 3.3

*Proof.* The proof will follow by application of a Fano type inequality on a well chosen set of problems.

**Step 1: Constructing our well chosen set of problems**    We define a set of grid points $U_0, U_1, ...$ recursively as follows, $U_0 = \arg\min(\Delta_i)$, for $m \geq 0$ we then define

$$U_{m+1} = \arg\min\{\Delta_i : \forall j \leq m, |\mu_{U_j} - \mu_i| \geq 3\Delta_i + 3\Delta_{U_j}\} .$$

Let $M$ be the largest integer for which $U_M$ exists. Note that the sequence $(\Delta_{U_m})_{m>M}$ is monotonically increasing and furthermore, for all $i \in [K]$,

$$|\{m \in [M] : |3\mu_{U_m} - 3\mu_i| \geq \Delta_{U_m} + \Delta_i\}| \leq 2 . \tag{22}$$

We then define the corresponding set of groups, $D_0, D_1, ..., D_M$ as follows. For $i \in [K]$ let $m, n \in [M]$ be such that, $\mu_{U_m} \leq \mu_i \leq \mu_{U_n}$ then set $i^+ = m \vee n$ and $i^- = m \wedge n$. If $|\mu_i - \mu_{i^*}| \leq \Delta_i + \Delta_{i^+}$ then $i \in D_{i^+}$, otherwise $i \in D_{i^-}$.

**Proposition B.1.** *For all $m \in [M]$ we have that, $\forall i \in D_m, \Delta_{U_m} \leq \Delta_i$.*

*Proof.* W.l.o.g assume $\mu_i \geq \mu_{U_m}$. Let $j$ be such that $|\mu_{U_j} - \mu_i| \leq 3\Delta_i + 3\Delta_{U_j}$. Via Equation (22) we have that $\mu_{U_m} \leq \mu_i \leq \mu_{U_j}$ and by definition of $D_m, D_j, m > j$. Thus $\nexists j < m : |\mu_i - \mu_{U_j}| \leq 3\Delta_{U_j} + 3\Delta_i$ and therefore, if $\Delta_i < \Delta_{U_m}$, then

$$\arg\min\{\Delta_i : \forall j \leq m - 1, |\mu_{U_j} - \mu_i| \geq 3\Delta_m + 3\Delta_j\} \neq U_m$$

which is a contradiction via the definition of $U_m$.

$\square$

For $m \in [M]$, set $W_m = \{i : |\mu_i - \mu_{U_m}| \leq 3\Delta_{U_m}\}$.

**Proposition B.2.** *For all $m \in [M]$,*

$$|\{i \in D_m : |\mu_i - \mu_{U_m}| \geq 3\Delta_{U_m}\}| \leq 36|W_m|$$

*Proof.* Firstly, by definition we have that,

$$\Delta_{U_m} \geq \frac{K\varepsilon p(1 - \mu_{U_m})}{|W_m|} . \tag{23}$$

Now, let $i^* = \arg\max_{i \in D_m \setminus W_m}(\Delta_i)$. As we have that $\forall i \in D_m, |\mu_i - \mu_{U_m}| \leq 3\Delta_i + 3\Delta_{U_m}$, the following holds,

$$\forall i \in D_m \setminus W_m, |\mu_i - \mu_{i^*}| \leq 3\Delta_{i^*} + 3\Delta_{U_m} .$$

Now consider the set of adjacent grid points $L \subset D_m \setminus W_m$ such that $\lambda\left(\bigcup_{i \in L}[G_i, G_{i+1}]\right) \leq \Delta_{i^*}$, for $j \in L$,

$$\Delta_j \leq \frac{K\varepsilon p(1 - \mu_j)}{|L|} \tag{24}$$

$$\leq \frac{K\varepsilon p((1 - \mu_{U_m}) + 3\Delta_j + 3\Delta_{U_m})}{|L|} \tag{25}$$

$$\leq \frac{K\varepsilon p(1 - \mu_{U_m})}{|L|} + \frac{6\Delta_j}{|L|} \tag{26}$$

Where the final inequality comes from the fact we assume $K\varepsilon p < 1$. Thus if we assume $|L| \geq 12|W_m|$, we have that, $\Delta_j \leq \frac{K\varepsilon p(1 - \mu_{U_m})}{2|C_k|}$, a contradiction via proposition B.1 and Equation (23)

$\square$

The proof of the following proposition follows via the same argument as in the proof of Proposition B.2.

**Proposition B.3.** *For all $m \in [M]$,*

$$|W_m| \leq 3|\{i \in W_m : |\mu_i - \mu_{U_m}| \leq \Delta_{U_m}\}|$$

We are now ready to construct our set of problems. Consider a family of problems $\nu^Q$ indexed by $Q \in \{0, 1\}^K$, where the target function $\eta_Q$ corresponding to $\nu^Q$ is defined as follows. Set the coefficients,

$$(\beta_m^1, \beta_m^2)_{m \in [M]}$$

and for $m \in [M]$, set $\beta_1^m = \mu_{U_m} + 4/3\Delta_{U_m}, \beta_2 = \mu_{U_m} - 4/3\Delta_{U_m}$, also define the sets

$$C_{m,1}^Q = \bigcup_{i \in W_m : Q_i = 1} [G_i, G_{i+1}) \qquad C_{m,2}^Q = \bigcup_{i \in W_m : Q_i = -1} [G_i, G_{i+1}) .$$

and $C_m = \bigcup_{i \in W_m} [G_i, G_{i+1})$ we then have

$$\eta_Q(x) = \sum_{m \in [M]} \mathbf{1}(x \in C_{m,1}^Q)\beta_m^1 + \mathbf{1}(x \in C_{m,2}^Q)\beta_m^2 .$$

Finally let $C_0^Q = [0,1] \setminus \{C_{m,1}^Q, C_{m,2}^Q : m \in [M]\}$ and note that for all $x \in C_0^Q$, $\eta(x) = 0$. The following Lemma shows that, for a problem $\nu \in \mathcal{B}$, the gaps and complexity across our family problems, $\nu^Q$ indexed by $Q$, does not differ too much from $\nu$.

**Lemma B.4.** *Let $Q \in \{0,1\}^K$, for all $m \in [M], i : [G_i, G_{i+1}) \in C_m$, we have that $\Delta_{U_m}/4 < \Delta_i^{(Q)} < 3\Delta_{U_m}$, where $\Delta_i^{(Q)}$ is the gap of point $i$ on problem $\nu^Q$*

*Proof.* We have immediately that $3\Delta_{U_m} \leq 3\Delta_{U_m}$, definition of $\Delta_{U_m}$ Then, by proposition B.3,

$$\Delta_{U_m}/3 \leq \frac{K\varepsilon p(1-\mu_{U_m})}{3|\{i : |\mu_i - \mu_{U_m}| \leq \Delta_\}|} \leq \frac{K\varepsilon p(1-\mu_{U_m})}{|W_m|} \leq \frac{K\varepsilon p(1-\mu_{U_m} \smile 4\Delta_{U_m}/3)}{|W_m|} + \frac{4K\varepsilon p\Delta_{U_m}}{3|W_m|},$$

which implies, in combination with our assumption $K\varepsilon p \leq 1/8$ that,

$$\Delta_{U_m}/6 \leq \frac{K\varepsilon p(1-\mu_{U_m} \smile 4\Delta_{U_m}/3)}{|W_m|}$$

and thus that for all $i \in W_m$, $\Delta_i^{(Q)} \geq \Delta_{U_m}/6$. $\qquad\square$

**Lemma B.5.** *Given, $\nu \in \mathcal{B}$, we have that for all $\nu^Q$, $i \in [K], \Delta_i^{(Q)} \geq \Delta_i/2$, where $\Delta_i^{(Q)}$ is the gap of point $i$ on problem $\nu^Q$. Furthermore, $\sum H_{i,Q}^{(1)} \geq c \sum H_i^{(1)}$ where $H_{i,Q}^{(1)}$ is the complexity of point $i$ on problem $\nu^Q$, for some absolute constant $c > 0$.*

*Proof.* We prove the first statement. For all $i : [G_i, G_{i+1}) \in C_0$ we have that $\Delta_i^{(Q)} = 1$ and otherwise the statement follows from Lemma B.5. The second statement then follows by combination of Lemma B.5 and Propositions B.2 and B.1.

$\qquad\square$

**Step 2: showing that one suffers $\varepsilon$ regret on a well chosen event**   Let $Q^i$ be the transformation of $Q$ that flips the $i$th coordinate,

$$Q_a^k = \begin{cases} Q_a \text{ If } a \neq k, \\ 1 - Q_a \text{ If } a = k. \end{cases}$$

We remind the reader that we denote the scoring function outputted by the learner as $\hat{s}$. Now define,

$$z_m := \min\left( z : H_{s_{\hat{s}}}(z) \geq \frac{\lambda\left(\bigcup_{n=1}^{m-1} C_n \cup C_{m,1}^Q\right)\left(1 - \kappa\left(\bigcup_{n=1}^{m-1} C_n \cup C_{m,1}^Q\right)\right)}{(1-p)} \right),$$

define the event,

$$\xi_{i,m} := \left\{ \{x \in [G_i, G_{i+1}) : \hat{s}(x) > z_m\} \leq \frac{1}{2K} \right\}.$$

and then the events,

$$\mathcal{E}_1^m := \left\{ \sum_{i:[G_i,G_{i+1})\in C_m, Q_i=1} \mathbf{1}(\xi_{i,m}) \leq \frac{3K\lambda(C_m)}{4} \right\}, \qquad \mathcal{E}_0^m := \left\{ \sum_{i:[G_i,G_{i+1})\in C_m, Q_i=0} \mathbf{1}(\xi_{i,m}) \geq \frac{K\lambda(C_m)}{4} \right\}.$$

For a set $C \subset [0,1]$, define,

$$\kappa(C) := \frac{1}{\lambda(C)} \int_C \eta(x) \, dx.$$

Let $\hat{Z}_m \subset [0,1]$ be the largest set such that, $\forall x \in \hat{Z}_m, y \notin \hat{Z}_m, \hat{s}(x) \leq \hat{s}(y)$, and,

$$\lambda(\hat{Z}_m)(1 - \kappa(\hat{Z}_m)) \leq \lambda\left(\bigcup_{n=1}^{m-1} C_n \cup C_{m,1}\right)\left(1 - \kappa\left(\bigcup_{n=1}^{m-1} C_n \cup C_{m,1}\right)\right).$$

Note that $\hat{Z}_m$ is not necessarily unique, in this case we choose an arbitrary such $\hat{Z}_m$. Furthermore define,

$$\hat{Z}_m^0 = \left\{ x \in \hat{Z}_m : x \in \bigcup_{n=1}^{m-1} C_n \right\}, \qquad \hat{Z}_m^1 = \left\{ x \in \hat{Z}_m : x \in C_{m,1} \right\},$$

and

$$\hat{Z}_m^2 = \left\{ x \in \hat{Z}_m : x \in C_{m,2} \cup \bigcup_{n=m+1}^{M} C_n \right\}.$$

And let $\hat{G}_m = \bigcup_{n=1}^{m-1} C_n \setminus \hat{Z}_m^0$. Note that under event $\mathcal{E}_1^m$, we have

$$\lambda\big(Z_m^1\big) \leq K\lambda(C_m)/4 . \tag{27}$$

Now, via definition of $\hat{Z}_m$, we have the following,

$$\lambda(\hat{Z}_m^1)(1-\kappa(C_{m,1}))+\lambda(\hat{Z}_m^2)(1-\kappa(C_{m,2})) \leq \lambda(C_{m,1})(1-\kappa(C_{m,1}))+\lambda(\hat{G}_m)(1-\kappa(\hat{G}_m)) , \tag{28}$$

Thus, for a problem $\nu^Q : \sum_{i:[G_i,G_{i+1}]\in C_m} Q_i \geq K\lambda(C_m)/2$, on event $\mathcal{E}_1^m$, via combination of Equations (27) and (28),

$$\lambda(\hat{Z}_m^2)(1 - \kappa(C_{m,2})) \leq 3\lambda(C_{m,1})(1 - \kappa(C_{m,1}))/4 + \lambda(\hat{G}_m)(1 - \kappa(\hat{G}_m)) \tag{29}$$

$$\leq (3\lambda(C_{m,1})/4 + \lambda(\hat{G}_m))(1 - \kappa(C_{m,1})) , \tag{30}$$

where the final inequality comes from the fact $\kappa(C_{m,1}) \geq \kappa(\hat{G}_m)$. To complete Step: 2 we now lower bound $d_\infty(\hat{s}, \eta)$ on event $\mathcal{E}_1^m$. Firstly note that,

$$\mathrm{ROC}\left( \frac{(1 - \hat{Z}_m)\lambda(\hat{Z}_m)}{1 - p}, \eta \right) = \frac{\lambda\left(\bigcup_{n=1}^{m-1} C_n \cup C_{m,1}\right)\kappa\left(\bigcup_{n=1}^{m-1} C_n \cup C_{m,1}\right)}{p}$$

$$= \frac{\lambda\left(\bigcup_{n=1}^{m-1} C_n\right)\kappa\left(\bigcup_{n=1}^{m-1} C_n\right)}{p} + \frac{\lambda(C_{m,1})\kappa(C_{m,1})}{p}$$

therefore, for a problem $\nu^Q : \sum_{i:P_i\in C_m} Q_i \geq K\lambda(C_m)/2$, on event $\mathcal{E}_1^m$,

$$d_\infty(\hat{s}, \eta) \geq \frac{\lambda(\hat{G}_m)\kappa(\hat{G}_m)}{p} + \frac{\lambda(C_{m,1})\kappa(C_{m,1})}{p} - \frac{\kappa(C_{m,1})\lambda(\hat{Z}_m^1)}{p} - \frac{\kappa(C_{m,2})\lambda(\hat{Z}_m^2)}{p}$$

$$\geq \frac{\lambda(\hat{G}_m)\kappa(\hat{G}_m)}{p} + \frac{3\lambda(C_{m,1})\kappa(C_{m,1})}{4p} - \frac{\kappa(C_{m,2})\lambda(\hat{Z}_m^2)}{p}$$

$$\geq \frac{\lambda(\hat{G}_m)\kappa(\hat{G}_m)}{p} + \frac{3\lambda(C_{m,1})\kappa(C_{m,1})}{4p} - \frac{\kappa(C_{m,2})(3\lambda(C_{m,1}/4 + \lambda(\hat{G}_m))}{p}\frac{1 - \kappa(C_{m,1})}{1 - \kappa(C_{m,2})}$$

$$\geq \frac{3\lambda(C_{m,1})}{4p}\left(\kappa(C_{m,1}) - \kappa(C_{m,1})\frac{1 - \kappa(C_{m,1})}{1 - \kappa(C_{m,2})}\right)$$

$$= \frac{3\lambda(C_{m,1})}{4p}\left(\frac{\kappa(C_{m,1}) - \kappa(C_{m,2})}{1 - \kappa(C_{m,2})}\right)$$

$$= \frac{3\lambda(C_{m,1})}{4p}\left(\frac{8/3\Delta_{U_m}}{1 - \kappa(C_{m,2})}\right) \geq \frac{3\lambda(C_{m,1} \cup C_{m,2})}{8p}\left(\frac{8/3\Delta_{U_m}}{1 - \kappa(C_{m,2})}\right) \geq \varepsilon$$

where the first inequality follows from Equation (27) and the second from (29).

Thus, as we assume policy $\pi$ is PAC$(\delta, \varepsilon)$, on all problems $\nu^Q$, we must have that, on all problems $\nu^Q : \sum_{i:[G_i,G_{i+1}]\in C_m} Q_i \geq K\lambda(C_m)/2$,

$$\mathbb{P}_{\nu^Q,\pi}(\mathcal{E}_1^m) \leq \delta , \tag{31}$$

Via similar reasoning we can show that on all problems $\nu^Q : \sum_{i:[G_i,G_{i+1}]\in C_m} Q_i \leq K\lambda(C_m)/2$, we must have that,

$$\mathbb{P}_{\nu^Q,\pi}(\mathcal{E}_0^m) \leq \delta . \tag{32}$$

**Step 4: bounding the probability of the sum of $\xi_i^m$** Now, for $m \in [M]$, via the Azuma hoeffding inequality applied to the martingale,

$$\sum_{i:[G_i,G_{i+1})\in C_m, Q_i=0} \left[\mathbf{1}(\xi_{i,m}) - \mathbb{P}_Q(\xi_{i,m})\right],$$

we have that,

$$\mathbb{P}_Q\left(\sum_{i:[G_i,G_{i+1})\in C_m, Q_i=0} \left[\mathbf{1}(\xi_{i,m}) - \mathbb{P}_Q(\xi_{i,m})\right] \geq K\lambda(C_m)\log\left(\frac{1}{1-\delta}\right)\right) \leq 1-\delta. \qquad (33)$$

Thus via combination of Equations (32) and (33) we must have that, $\forall Q : \sum_{i:[G_i,G_{i+1})\in C_m}^K Q_i \leq K\lambda(C_m)/2$,

$$\sum_{i:[G_i,G_{i+1})\in C_m, Q_i=0} \mathbb{P}_Q(\xi_{i,m}) \leq \lambda(C_m)\left(\frac{K}{4} + K\log\left(\frac{1}{1-\delta}\right)\right) \leq \frac{3K\lambda(C_m)}{8}, \qquad (34)$$

where the second inequality comes from our assumption $\delta \leq 1 - \exp(-1/8)$. Via similar reasoning we also have that, $\forall Q : \sum_{i:[G_i,G_{i+1})\in C_m} Q_a \geq K\lambda(C_m)/2$

$$\sum_{i:[G_i,G_{i+1})\in C_m, Q_i=1} \mathbb{P}_Q(\xi_{i,m}) \geq \lambda(C_m)\left(\frac{K}{2} - K\log\left(\frac{1}{1-\delta}\right)\right) \geq \frac{5K\lambda(C_m)}{8}. \qquad (35)$$

**Step 5: applying a Fano type inequality** We first define the class of problems upon which we will apply Fano.

$$\mathfrak{Q} = \left\{Q : \forall m \in [M], \sum_{i:[G_i,G_{i+1})\in C_m} Q_i = \frac{K\lambda(C_m)}{2}\right\}$$

$$\mathfrak{Q}_0 = \left\{Q : \forall m \in [M], \sum_{i:[G_i,G_{i+1})\in C_m} Q_i = \frac{K\lambda(C_m)}{2} - 1\right\}$$

We remind the reader that for $i \in [K]$, we write $\tau_i = \sum_{t=1}^\tau \mathbf{1}(a_t \in [G_i,G_{i+1}))$ and for $m \in [M]$, $\tau_{(m)} = \sum_{i:[G_i,G_{i+1})\in C_m} T_i$. We see that, for all $Q \in \mathfrak{Q}_0, i : Q_i = 0$, there exists a unique $\tilde{Q} \in \mathfrak{Q}$ such that $\tilde{Q}^i = Q$, therefore,

$$\sum_{Q\in\mathfrak{Q}}\sum_{m\in[M]}\sum_{i:[G_i,G_{i+1})\in C_m, Q_i=1} \mathbb{P}_{Q^i}(\xi_{i,m}) = \sum_{Q\in\mathfrak{Q}_0}\sum_{m\in[M]}\sum_{i:[G_i,G_{i+1})\in C_m, Q_i=0} \mathbb{P}_Q(\xi_{i,m}),$$

and thus, using the data processing inequality and the convexity of the relative entropy we have,

$$\mathrm{kl}\left(\underbrace{\frac{1}{|\mathfrak{Q}|}\sum_{Q\in\mathfrak{Q}}\sum_{m\in[M]}\frac{2}{K\lambda(C_m)}\sum_{i:P_i\in C_m, Q_i=1}\mathbb{P}_{Q^i}(\xi_{i,m})}_{\leq 6/8}, \underbrace{\frac{1}{|\mathfrak{Q}|}\sum_{Q\in\mathfrak{Q}}\sum_{m\in[M]}\frac{2}{K\lambda(C_m)}\sum_{i:P_i\in C_m, Q_i=1}\mathbb{P}_Q(\xi_{i,m})}_{\geq 10/8}\right)$$

$$\leq \frac{1}{|\mathfrak{Q}|}\sum_{Q\in\mathfrak{Q}}\sum_{m\in[M]}\frac{2}{K\lambda(C_m)}\sum_{i:P_i\in C_m, Q_i=1}\mathbb{E}_Q[\tau_m]\frac{\mathrm{kl}(\mu_{U_m}-4/3\Delta_{U_m}, \mu_{U_m}+4/3\Delta_{U_m})}{2}$$

$$\leq \frac{1}{|\mathfrak{Q}|}\sum_{Q\in\mathfrak{Q}}\sum_{m\in[M]}\frac{\mathbb{E}_Q[\tau_{(m)}]\,\mathrm{kl}(\mu_{U_m}-4/3\Delta_{U_m}, \mu_{U_m}+4/3\Delta_{U_m})}{K\lambda(C_m)}$$

$$\leq \max_{Q\in\mathfrak{Q}}\sum_{m\in[M]}\frac{\mathbb{E}_Q[\tau_{(m)}]\,\mathrm{kl}(\mu_{U_m}-4/3\Delta_{U_m}, \mu_{U_m}+4/3\Delta_{U_m})}{K\lambda(C_m)}.$$

Then using the Pinsker inequality $\mathrm{kl}(x, y) \geq 2(x - y)^2$, we obtain

$$\frac{1}{|\mathfrak{Q}_m|} \sum_{Q \in \mathfrak{Q}_m} \sum_{m \in [M]} \frac{2}{K\lambda(C_m)} \sum_{i:P_i \in C_m, :Q_i=1} \mathbb{P}_Q(\xi_{i,m}) \leq \frac{3}{8} + \sqrt{\max_{Q \in \mathfrak{Q}} \sum_{m \in [M]} \frac{\mathbb{E}_Q[\tau_{(m)}] \mathrm{kl}(\mu_{U_m} - 4/3\Delta_{U_m}, \mu_{U_m} + 4/3\Delta_{U_m})}{K\lambda(C_m)}}$$

and therefore,

$$\max_{Q \in \mathfrak{Q}} \sum_{m \in [M]} \frac{\mathbb{E}_Q[\tau_{(m)}] \mathrm{kl}(\mu_{U_m} - \Delta_m, \mu_{U_m} + \Delta_m)}{K\lambda(C_m)} \geq \frac{9}{64}$$

thus

$$\max_{Q \in \mathfrak{Q}} \sum_{m \in [M]} \mathbb{E}_Q[\tau_{(m)}] \geq c' \sum_{m \in [M]} \frac{K\lambda(C_m)}{\mathrm{kl}(\mu_{U_m} - 4/3\Delta_{U_m}, \mu_{U_m} + 4/3\Delta_{U_m})}$$

where $c' > 0$ is an absolute constant. The proof now follows, as $\forall m \in [M], i : [G_i, G_{i+1}) \subset C_m$, $H_i \geq \frac{1}{\mathrm{kl}(\mu_{U_m} - 4/3\Delta_{U_m}, \mu_{U_m} + 4/3\Delta_{U_m})}$. $\qquad\square$

