# OpenReview forum: "Active Bipartite Ranking"
_NeurIPS.cc/2023/Conference — NeurIPS 2023 poster_

### Official Review · Reviewer_jBg3 · 2023-07-02

**Soundness:** 3 good
**Presentation:** 3 good
**Contribution:** 3 good
**Rating:** 7
**Confidence:** 2

**Summary:**

The paper proposes an elimination active learning criterion for the bipartite ranking problem. The criterion account for the local smoothness of a data sample’s posterior distribution. The author proves the sample complexity and the PAC learnability of the proposed algorithm under some assumptions.


**Strengths:**

1The proposed methodology is theoretically sound. The removal of a point in the active set corresponds to the learning objective, which is to minimize the gap between the current ROC and the optimal ROC. Beyond that, the PAC learning bound has been derived.

**Weaknesses:**

1The proposed method seems not to be practically sound. Most learning problems nowadays have high input dimensions. It is not clear whether the proposed algorithm can be easily extended to solve those real-world problems or if it only enjoys the nice properties for one dimension data.
2It is not clear how the criterion will change after the removal of a data point. If the impact on the posterior around data point i is dependent on other points, is it still a fair criterion?


**Questions:**

My questions are listed in the weakness section.

---

> ### Author Response · Authors · 2023-08-12
>
> Please see the general rebuttal in regards to your first question.
>
> Indeed, due to the interconnected nature of the problem the decision to remove a point may depend on points already removed from the active set, a problem not found in multi armed bandits, see the discussion at the end of section 3.1. For this reason a point is removed from the active set only when the number of points appearing within a certain distance of it is sufficiently small. This distance is $\Delta_{(t)}$, multiplied by some large constant. Thus any points $j$ not yet clearly distinguished from the removed point also has the guarantee that $\Delta_j \leq c\Delta_{(t)}$, for some constant $c$, thus ensuring that our criterion is consistent. See the proof of Lemma A.3 for details.

---

> > ### Comment · Reviewer_jBg3 · 2023-08-15
> > **Thank you for your response**
> >
> > My concerns have been addressed, and I have changed my rating accordingly.

---

### Official Review · Reviewer_rRJf · 2023-07-06

**Soundness:** 3 good
**Presentation:** 2 fair
**Contribution:** 2 fair
**Rating:** 4
**Confidence:** 1

**Summary:**

The paper proposes an active learning for the bipartite ranking problem in which the goal is to provide a ranking for a set of points rather than given positive or negative labels. The proposed algorithm called active-rank tries to optimize the ROC curve, evaluated by the distance to the optimal one. The authors provided detailed theoretical analysis that guarantees the distance can be arbitrary small with high probability. The evaluation is performed on synthetic data.

**Strengths:**

- The paper appears to be technically sound, and the theoretical analysis is in depth.

- According to the authors, the paper is the first rigorous framework for active bipartite ranking.

**Weaknesses:**

- The paper is difficult to follow for those who are not familiar with this topic.

- The dimension of the input is restricted to 1, for which significance is not clear to me. The authors should have clarified it more.

- The experiments are only for synthetic data.

**Questions:**

- The authors assumes the piecewise constant model. Is it a reasonable assumption for the bipartite ranking setting?

- K is assumed to be known. How is it selected in practice?

- In section 4, scenario 1 is 'K = 16'. Is it 'K = 4'?

**Limitations:**

Some limitations are discussed in the conclusion section.

---

> ### Author Response · Authors · 2023-08-12
>
> Please see the general rebuttal in regards to your first  and second questions.
>
> The definition of scenario 1 was wrong, the correct version is,
>
> Scenario 1: $\mu_i = 0$ when $i\leq 11$, $\mu_{12}=\mu_{13} = 0.23$, $\mu_{14}=\mu_{15} = 0.33$, $\mu_{16}=0.35$    and $K = 16$.
>
> This scenario aims at testing the algorithm when a significant section of the feature space is constant.

---

### Official Review · Reviewer_RSb9 · 2023-07-10

**Soundness:** 3 good
**Presentation:** 2 fair
**Contribution:** 3 good
**Rating:** 7
**Confidence:** 2

**Summary:**

This paper develops an active learning framework for the bipartite ranking problem. A selective sampling procedure or strategy is necessary for lebeling data points sequentially. The proposed method is called active-rank, which aims to minimize the distance between the ROC curve of the ranking function built and the optimal one, w.r.t., the sup norm. Theoretical results as well as some associated numerical results based on synthetic data demonstrate strong empirical evidence of performance.


**Strengths:**

- The paper is well-written with clear notation.
- Concrete theoretical results are provided and discussed for the proposed algorithm.
- The performance of the proposed active-rank method is demonstrated by experiments that are in line with the theoretical considerations in the theory and active sampling procedure.

**Weaknesses:**

- Given the elimination nature of the proposed method, the experiment lacks behavioral observations of the proposed method in real dataset, that may be helpful for the practitioner's reference.

**Questions:**

- Can you describe the pros and cons for practitioners, when working on real datasets where the scenario situation described in Section 4 is not clear?

**Limitations:**

no concern

---

### Official Review · Reviewer_WRWn · 2023-07-12

**Soundness:** 3 good
**Presentation:** 3 good
**Contribution:** 2 fair
**Rating:** 6
**Confidence:** 3

**Summary:**

An active bipartite ranking algorithm is proposed in this article for one-dimensional data with a posterior probability $\eta(x) = {\rm P}(Y =+1\vert X =x )$ piecewise constant on a grid of size $K$. An upper bound is provided on the number of queried to achieve a controlled error of type PAC$(\epsilon,\delta)$ on the ${\rm sup}$ norm between the learned ROC curve and the optimal one. Compared to the lower bound of any sampling policy also established in this paper, the upper bound suffers notably from a logarithmic dependence on the size $K$ of the grid. Experiments on synthetic data conforming to the aforementioned setting show a consistent superiority of the proposed algorithm over several baseline. However a really close match is observed between the uniform passive sampling and the proposed active method.

**Strengths:**

* The studied problem, active bipartite ranking, is practically interesting and original.
* The theoretical framework is clearly presented with a brief explanation of its relation to the best arm identification (BAI) problem.
* Statistical guarantees are provided to support the proposed algorithm and to lay the ground for future investigation.
* The proposed algorithm is tested on synthetic data.

**Weaknesses:**

* The data setting is quite restrictive in a one-dimensional space with the posterior probability piecewise constant on a grid of a known size $K$.
* The passive sampling performs comparably to the proposed active algorithm under various scenarios conforming to the data setting underlying the proposed algorithm.
* There is no testing on real data.

**Questions:**

* Does there exist an upper bound of the uniform passive sampling under the same setting or a similar one? Seeing the empirical results, I suspect that the passive sampling achieves already nearly optimal performance, possibly due to the fact that the posterior is piecewise constant on a uniform partition of the space and data points inside each interval are statistically equivalent.

**Limitations:**

The authors have discussed the extensions to smooth posteriors and to higher dimensions.

---

> ### Comment · Reviewer_WRWn · 2023-08-20
>
> I thank the authors for the discussion on the comparison to passive learning, which is helpful and worth developing in the main text. My score is increased.

---

### Official Review · Reviewer_iq8o · 2023-07-26

**Soundness:** 3 good
**Presentation:** 2 fair
**Contribution:** 2 fair
**Rating:** 6
**Confidence:** 4

**Summary:**

- This paper studies sample complexity bounds for bipartite ranking under an active learning paradigm. The paper gives an algorithm called “active-rank” to solve the variant of the problem under the assumption that the posterior is a piecewise constant function over a grid of size K. The sample complexity bounds are problem-dependent, for which the authors define an instance-dependent notion of problem complexity where each arm/grid point’s global effect needs to be treated individually.

- The paper also gives a lower bound (not matching with the upper bound), for the class of problem instances where the posterior differs only by a constant over the input range. The authors hope that the dependency of these bounds on $\Delta$ and the logarithmic gap in the upper bound can be removed but leave it as a future work.

**Strengths:**

- Although not new, active learning is an important learning paradigm, and its application to bipartite ranking is a significant contribution.

- The ideas in the paper are well-presented. The discussion about the global nature of the problem compared to other works where only pairwise comparisons, a local property, needs to be estimated with high error tolerance and probability helped access the novelty of this work.

**Weaknesses:**

- The results in the paper hold only for a carefully designed set of problem instances of bipartite ranking. However, the results contribute toward improving the understanding of active learning in bipartite ranking. So I wouldn’t consider this a weakness.

- However, a theoretical comparison with passive ranking complexity is missing. Are there no existing works studying this? It is necessary to understand how the sample complexity compares between active and passive learning. What does uniform sampling give on the grid? Does active learning give any improvement over passive learning? The authors say before Assumption 2.1 that the assumption helps in understanding the power of active sampling over passive, but I did not see anywhere in the paper a theoretical comparison.

- The presentation of the paper can be improved. The figures are hard to read; the axes are not labelled (Fig 4) and sometimes the legends are missing (Fig 2).

**Questions:**

- Please answer my question about the theoretical comparison of active vs. passive ranking for the 1D grid.

- The explanation about Lemma 2.2 does not seem correct, even after ignoring the typos in Lines 158 and 159. Delta_i is an upper bound on how big a confidence interval you can have around $\mu_i$. The inequality should be $|\mu_i - \mu_j| \ge \Delta_i$. Only for such $j \neq i$, the scoring function has to get the sign correct. Is that what you meant?

- Line 319 says, “active-rank all competitors”. What is missing? Active-rank outperforms all the competitors? That doesn’t seem true from Figure 3.

- Is the x-axis in Figure 4 stopping time?

**Limitations:**

Limitations adequately addressed

---

> ### Author Response · Authors · 2023-08-12
>
> Please see the general rebuttal in regards to your first question.
>
> Line 159 is a typo, it should read "for all $j: |\mu_i - \mu_k| \geq \Delta_i$".
>
> The x-axis in figure 4 is indeed stopping time, this will be clarified.
>
> Aside from scenario one, for larger stopping times, i.e. greater than 500 active-rank outperforms the competitors, although by an admittedly small margin in some cases. The typo will be corrected to say something to this effect.

---

> > ### Comment · Reviewer_iq8o · 2023-08-13
> > **Response to rebuttal**
> >
> > Thanks for the comparison with the passive approach. It helps judge the contribution of this work better. Increasing my score.

---

### Author Rebuttal · Authors · 2023-08-09

We thank all reviewers for the time taken to read our paper and for their constructive feedback. Some topics come up for several reviewers.

Performance of passive approach in comparison to our active algorithm:

For a uniform/passive sampling strategy to be PAC$(\epsilon,\delta)$ one would have to draw samples until the width of the confidence interval at all points $i$ is less than $\Delta_i$. Therefore, a uniform sampling strategy, with an appropriate stopping rule, would have the following tight upper bound on its expected sampling time, up to log terms, $cK\max_{i\in [K]}\frac{1}{kl(\mu_i,\mu_i + c'\Delta_i)},$ for some absolute constants $c,c'>0$. Essentially, our improvement in the active setting is to replace the $\max$ with a weighted summation across the grid. Thus, in settings where the $\Delta_i$ are relatively constant across large sections of the grid then the theoretical performance of a passive approach can be close to optimal. In contrast, in cases where a very small section of the interval is hard to rank and the rest is easy - i.e. $K\max_{i\in [K]}\frac{1}{kl(\mu_i,\mu_i + \Delta_i)}$ is much greater than $\sum_{i\in [K]}\frac{1}{kl(\mu_i,\mu_i + \Delta_i)},$ a passive approach will fail. Such settings involve large $K$  where the gaps $\Delta_i$ on the majority of cells are large, with a relatively small number of cells with small gaps $\Delta_i$. Incidentally, we point out that this corresponds to many situations of interest in practice (in information retrieval, for a specific request, the vast majority of the documents are equally irrelevant, while the ranking of a very small fraction of relevant documents is challenging; the same phenomenon is also observed in credit-risk screening). In these cases the benefit to the practitioner will then be that they quickly focus on the interesting sections of the feature space. For such a setting one needs very small tolerance $\epsilon$ as well as large $K$. This results in computational difficulties for experiments and as noted in the paper, some work is required to reveal a setting where the active rank show a large improvement over a uniform approach. We agree with the reviewers that comparison to a passive approach is lacking, a final version of the paper would include additional discussion in line with the above.

Extension to higher dimensions:

Our analysis extends immediately to higher dimensions. Indeed one may consider a function piece wise constant on a $d$ dimensional grid of size $K$ and as we can always project a $d$ dimensional grid onto the $[0,1]$ interval, our results remain unchanged. Indeed, for better illustration, our experiments are carried out in dimension 2. A more interesting question is to include a sparsity assumption with increasing dimension $d$, as this arises in many practical situations. This would fundamentally change the nature of the problem and is beyond the scope of this paper. A discussion will be added in the final version for the sake of clarity.

The piece wise constant assumption and assumed knowledge of $K$:

For active learning in general such an assumption is very reasonable, first and foremost as it is essential for the majority of the literature in finite multi armed bandits. Indeed our Bipartite ranking problem can be viewed as a finite multi armed bandit where the objective of the learner is to rank the $K$ arms. Other pure exploration problems, such as best arm identification, have seen extensive interest, in the case of fixed known $K$. We believe ranking the arms is as natural a question as finding the best arm and the potential for practical application is no less.

In practice one can replace knowledge of $K$ with an upper bound, although this will then appear in the bounds. Selecting $K$ is a tricky question, ideally one would have expert knowledge, however, in the absence of this it may be better to remove the piece wise constant assumption and view the problem under a continuity assumption, e.g. $\beta$-Hölder smoothness constraint. This is akin to the extension of the finite multi armed bandit to the infinite armed bandit. Now the question becomes, how does one proceed without knowledge of $\beta$? Ideas from the bandit literature may be applied to our setting, e.g. Carpentier, Valko, 2015, where $\beta$ is estimated. This question goes beyond the scope of our paper and will also be especially challenging in higher dimensions.

Reviewer 5:

Indeed, due to the interconnected nature of the problem the decision to remove a point may depend on points already removed from the active set, a problem not found in multi armed bandits, see the discussion at the end of section 3.1. For this reason a point is removed from the active set only when the number of points appearing within a certain distance of it is sufficiently small. This distance is $\Delta_{(t)}$, multiplied by some large constant. Thus any point $j$ not yet clearly distinguished from the removed point also has the guarantee that $\Delta_j \leq c\Delta_{(t)}$, for some constant $c$, thus ensuring that our criterion is consistent. See the proof of Lemma A.3 for details.

Reviewer 1:

Line 159 is a typo, it should read "for all $j: |\mu_i - \mu_k| \geq \Delta_i$". The x-axis in figure 4 is indeed stopping time, this will be clarified. Aside from scenario one, for larger stopping times, i.e. greater than 500 active-rank outperforms the competitors, although by an admittedly small margin in some cases. The typo will be corrected to say something to this effect.

Reviewer 4:

The definition of scenario 1 was wrong, the correct version is,

Scenario 1: $\mu_i = 0$ when $i\leq 11$, $\mu_{12}=\mu_{13} = 0.23$, $\mu_{14}=\mu_{15} = 0.33$, $\mu_{16}=0.35$    and $K = 16$.

This scenario aims at testing the algorithm when a significant section of the feature space is constant.

---

### Decision · Program_Chairs · 2023-09-21

**Decision:**

Accept (poster)

**Comment:**

The paper proposes Active rank, a new ranking algorithm under the Active learning framework, for bi-partite ranking. It provides rigorous bounds on sample complexity for this result. Overall this should be of interest to ranking community and hence to Neurips audience.